# Unlock the Black Box by Interpreting Graph Convolutional Networks via Additive Decomposition

## Abstract

The vast adoption of graph neural networks (GNNs) in broad applications calls for versatile interpretability tools so that a better understanding of the GNNs' intrinsic structures can be gained. We propose an interpretable GNN framework to decompose the prediction into the additive combination of node features' main effects and the contributions of edges. The key component of our framework is the generalized additive model with the graph convolutional network (GAM-GCN) that allows for global node feature interpretations. The inherent interpretability of GAM and the expressive power of GCN are preserved and naturally connected. Further, the effects of neighboring edges are measured by edge perturbation and surrogate linear modeling, and the most important subgraph can be selected. We evaluate the proposed approach using extensive experiments and show that it is a promising tool for interpreting GNNs in the attempt to unlock the black box.

## 1 Introduction

In the deep learning era, numerous neural network architectures have been proposed and successfully applied to real-world applications. The graph neural network (GNN) has drawn much attention in recent years (Sperduti & Starita, 1997; Kipf & Welling, 2017; Xu et al., 2018; Bresson & Laurent, 2017; Veličković et al., 2018). Various graph data can be directly modeled by GNNs for various tasks, e.g., medical imaging analysis (Liu et al., 2021), node classification in social networks and graph classification of chemical molecules (Zhou et al., 2020).

Recently, the machine learning community has witnessed an increasing number of GNN variants. Among them, the most popular one can be referred to as the graph convolutional network (GCN) (Kipf & Welling, 2017). Variants of GCN include the graph isomorphism network (GIN) (Xu et al., 2018), gated graph convolutional networks (GatedGCN) (Bresson & Laurent, 2017), graph attention networks (GAT) (Veličković et al., 2018), etc. A comprehensive comparison of different GNNs can be found in Dwivedi et al. (2020). Despite the increasingly powerful predictive performance, most state-of-the-art GNNs are designed to be a black box. The network architectures are becoming more and more complicated, while we only have very limited access to their inner decision-making processes.

It is a challenging and promising research direction to open the black box of GNNs. Some representative work can be referred to the gradient-based methods, e.g., gradient-weighted class activation mapping (Grad-CAM) (Pope et al., 2019); perturbation-based methods, e.g., GNNExplainer (Ying et al., 2019) and the parameterized explainer (PGExplainer) (Luo et al., 2020); decomposition-based methods, e.g., GNN with layer-wise relevance propagation (GNN-LRP) (Baldassarre & Azizpour, 2019); and surrogate models, e.g., GraphLime (Huang et al., 2020). For a comprehensive review of interpretable GNN techniques, see Yuan et al. (2020). In summary, the aforementioned methods are all post-hoc explanation tools. They target interpreting the fitted black-box models, while the provided explanations are typically approximations of the original GNNs, and hence may not fully capture the hidden patterns and intrinsic structures.

In this paper, we propose an interpretable graph convolutional network for node prediction tasks. Intuitively, the prediction can be disentangled into two parts, i.e., the main effects of node features and the contributions of its neighboring edges. A generalized additive model (GAM) architecture is proposed to help interpret

GCN, such that the pure main effects of node features can be efficiently quantified. Thus, we can obtain a comprehensive global understanding of how different features contribute to the predictions. As the nodes in a graph usually have varied neighboring topology, we need a fine-grained local explanation to understand the relationships between the target node and its corresponding neighbor structures. By post-hoc edge perturbation and surrogate linear modeling, the effects of neighboring edges can be measured for individual nodes. With the surrogate linear model, not only can the most important subgraph for the prediction be selected, but we can also quantify the positive or negative attributions of each edge. Experiments conducted on multiple synthetic and real datasets verify that the proposed approach can easily generate interpretable results, and the deduced interpretation serves as a good approximation to the fitted black-box model.

## 2 Related Work

### 2.1 Generalized Additive Models

In statistical machine learning, the generalized additive model (GAM) (Hastie & Tibshirani, 1990) is an intrinsically interpretable model that extends linear models with nonparametric ridge functions. In addition to conventional smoothing splines, the ridge functions can also be fitted by tree ensembles; see an empirical study of GAM in Lou et al. (2012). Recently, it has drawn extensive attention that GAM can also be reparametrized using neural network architectures (Agarwal et al., 2020), which shares the same idea as that in explainable neural networks (Yang et al., 2021a;b). Each ridge function is replaced by a subnetwork consisting of one input neuron, multiple hidden layers, and one output neuron, and the whole model can be jointly optimized through modern network training techniques.

### 2.2 Explainable GNNs

Interpretable machine learning is an emerging topic in both industry and academia. It aims at making the machine learning models easily understandable and interpretable, so that further improvements on GNNs can be achieved. In general, there are two types of interpretability, i.e., intrinsic interpretability and post-hoc interpretability. The former pursues intrinsically interpretable models while the predictive performance may be sacrificed, e.g., the shallow decision trees. The latter employs external tools to investigate a fitted black-box model, e.g., the local interpretable model-agnostic explanations (LIME) (Ribeiro et al., 2016), SHapley Additive exPlanations (SHAP) (Lundberg & Lee, 2017). The LIME and SHAP have been extensively used for analyzing models based on tabular data, while the research for GNN interpretability is a relatively new topic. Below, some representative strategies for interpreting GNNs are reviewed.

**Grad** (Gradient method) calculates the derivative of the final prediction with respect to each edge in the adjacency matrix (Pope et al., 2019). In general, it is assumed that edges with larger derivatives (in absolute values) tend to be more influential for the final prediction. Hence, the magnitude of gradients can be treated as the importance measure for the edges. This strategy is straightforward, and the gradients can be easily obtained. Hence, this method is commonly used as the baseline for edge attribution comparisons.

**GNNExplainer** is a post-hoc explanation tool that aims at selecting an important subgraph that best explains the final prediction (Ying et al., 2019). Edge masks are first introduced as trainable parameters and then optimized using the gradient descent to maximize the mutual information between the original and edge-masked predictions. As the optimization procedure terminates, a threshold of edge importance can be specified to select a compact subgraph that best explains the prediction.

**PGExplainer** is a parameterized method that can provide a global interpretation of GNNs (Luo et al., 2020). A multiple-layer perceptron (MLP) is introduced as the surrogate model to measure the importance of each edge in the graph. The surrogate MLP takes the embeddings of each edge as inputs and then outputs the mask probability of each edge. In particular, the edge embedding is formed by concatenating the embeddings of its two connecting nodes derived from the fitted GNN. The probabilities are then approximately discretized and input to the original GNN as edge masks. The MLP parameters are trained by maximizing the mutual information between the original predictions and the new ones generated by the edge-masked graph. Once the surrogate MLP is trained, the importance of each edge in the graph can be well quantified.

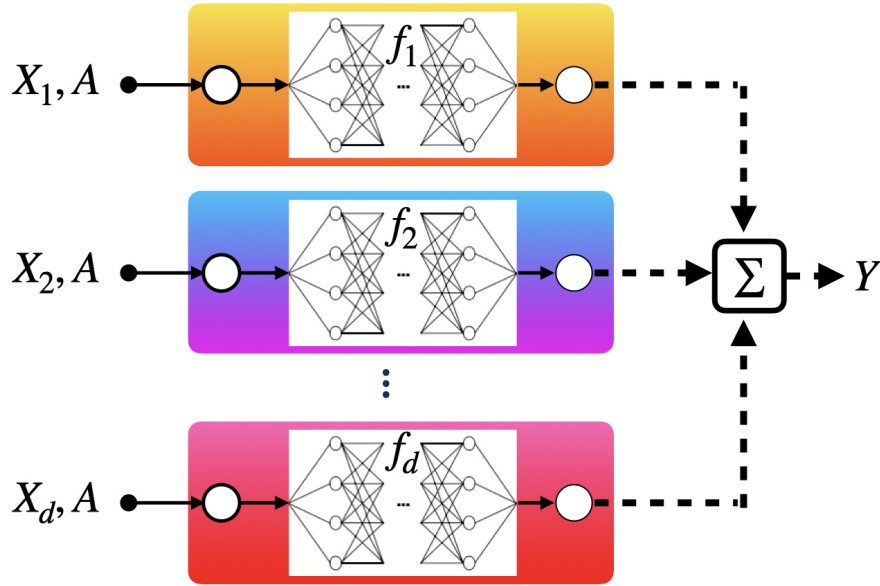

Figure 1: Architecture of the proposed GAM-GCN.

**DEGREE** decomposes the information flow in a trained GNN model to track the contributions of different components of the input graph to the final prediction. This allows estimating the importance of any node group without modifying the input (Feng et al., 2023). It designs specific decomposition schemes for commonly used GNN layers like graph convolution and graph attention layers. This enables isolating the information flow from different node groups. An aggregation algorithm is proposed to extend DEGREE to find important subgraphs instead of just nodes. It efficiently searches for subgraphs that maximally affect the prediction.

## 3 Methodology

Our goal is to develop a new interpretable GNN model by leveraging the well-studied GAM in statistics. The main idea is to decompose the prediction into the additive combination of two components, i.e., the contributions of node features and edges of the target prediction.

### 3.1 Notations

Let $G = (V, E)$ be an undirected graph that contains a set of nodes $V$ and a set of edges $E$. The node feature matrix is denoted as $X \in \mathbb{R}^{n \times d}$, where $n$ is the number of nodes and $d$ is the number of node features. The corresponding node response is represented by $Y$. For a node $v \in V$, its $K$-hop neighboring subgraph is defined as $G_v^K$, and the corresponding set of edges and set of nodes are denoted by $E_v^K$ and $V_v^K$. The number of edges in the subgraph is marked as $m$. In addition, the adjacency matrix is denoted by $A$. The adjacency matrix with self-connections is denoted as $\tilde{A} = A + I_n$ where $I_n$ is the identity matrix of dimension $n$, and the symmetrically normalized version $\hat{A} = \tilde{D}^{-\frac{1}{2}} \tilde{A} \tilde{D}^{-\frac{1}{2}}$ can be computed in a pre-processing step where $\tilde{D}$ denotes the corresponding degree matrix.

### 3.2 Global Interpretation for Node Features

The generalized additive model (GAM) takes the form of

$$g(\mathbb{E}(Y|X)) = \beta_0 + \sum_{j=1}^{d} h_j(X_j), \tag{1}$$

where $g$ is a link function, $\beta_0$ is the intercept, and $h_j$ is the ridge function for $j = 1, \ldots, d$. Each $h_j$ of the GAM can be re-parametrized by a modularized neural network which can be efficiently trained by modern deep learning techniques (Agarwal et al., 2020; Yang et al., 2021b). We propose a novel architecture, named GAM-GCN, which shares the same idea of GAM, as shown in Figure 1. The main difference is that in GAM-GCN each neuron is replaced by a GCN node. The GAM-GCN model takes the form of

$$g(\mathbb{E}(Y|X,A)) = \mu + \sum_{j=1}^{d} f_j(X_j, A), \tag{2}$$

where $X = (X_1, \ldots, X_d)$ is the node feature matrix with $d$ features, $A$ is the adjacency matrix, $\mu$ is the intercept, and $f_j$ is the feature network for the $j$-th node feature. Each $f_j$ contains one input GCN neuron, multiple hidden GCN layers, and one output GCN neuron. We use the ReLU activation function between GCN layers. Each node feature is connected with the final output by a feature network $f_j$. The main effects learned by $f_j$ are linearly combined to form the final output. To handle the multi-output tasks, such as multi-class node classification, the number of output neurons for each $f_j$ is the number of the final outputs. Thus, the corresponding contributions for features are linearly combined within each output dimension. The prediction on the $k$-th output dimension is given by

$$g(\mathbb{E}(Y^k|X,A)) = \mu^k + \sum_{j=1}^{d} f_j^k(X_j, A), \tag{3}$$

where $\mu^k$ is the intercept on the $k$-th dimension and $f_j^k$ is the contribution of the $k$-th output neuron of $f_j$.

Like other GNN models, the loss of GAM-GCN is chosen according to the tasks; for example, we use the cross-entropy loss for node classification. During the training, the following regularization techniques are applied to avoid overfitting. Within each feature network, the weights are dropped out and their $l_2$ norm is penalized. Also, we drop out individual feature networks, and the $l_2$ norm of the output of each feature network is penalized. Once the model training using backpropagation is finished, the pure main contributions of node features can be quantified when we remove all the edges. The overall importance (OI) of each node feature can be quantitatively measured by the mean absolute score as follows,

$$\text{OI}(j) = \frac{\sum_{v \in V} |f_j(X_j, I_n)[v] - b_j|}{nT}, \tag{4}$$

where $T = \sum_{j=1}^{d} \text{OI}(j)$ and $b_j = \sum_{v \in V} f_j(X_j, I_n)[v]/n$. The OI's of all the features are summed up to one, and the features with larger OI values are more important for the decision making process of GAM-GCN. The main effects, the shape functions learned by the feature networks for each node feature, can be easily visualized by plotting $f_i$ against $X_i$ using 2D line plots (for continuous or ordinal variables) or bar plots (for categorical variables). Apart from serving as the explanation, these plots also reflect the decision making process of GAM-GCN.

### 3.3 Local Interpretation for Neighboring Edges

For a node $v \in V$, the global main effects of node features can be investigated once the proposed GAM-GCN is well trained. However, examination of the effects of each node's neighboring edges are also non-trivial. Without quantifying their contributions, we may not be able to properly explain the final prediction. To measure the effects of node $v$'s neighboring edges, we first refine the exploration space to its $K$-hop neighboring subgraph $G_v^K$. The value of $K$ can be determined according to the architecture of the underlying GNN model. For instance, $K$ is set to be 3 if the GNN to be explained has three graph convolutional layers.

#### 3.3.1 Edge Attribution

The attribution of each edge can be measured following the steps below. First, we randomly remove some edges in the $K$-hop subgraph, and obtain the vector $Z^l = \{z_j^l\}_{j \in [M]}$ and $Z^r = \{z_j^r\}_{j \in [m]}$, where $[m]$ is the set of edges included in the $K$-hop subgraph. For undirected graphs, we find it is beneficial to treat the

---

**Algorithm 1:** The proposed edge attribution algorithm

---

**Input:** $v$ (Target node), $N$ (Number of resampling points), $\lambda$ (Average number of edges to be removed), $K$ (Number of hops to select neighboring edges)

**Output:** The selected subgraph and linear coefficients.

**1** Obtain the $K$-hop neighboring subgraph $G_v^K$ of the target node $v$;

**2 for** $i = 1,\ldots, N$ **do**

**3**     Randomly generate $L$ (cliped by $2M$) from a Poisson distribution with mean $\lambda$;

**4**     Randomly remove $L$ edges in $G_v^K$, and form the binary vectors $Z_i^l$ and $Z_i^r$;

**5**     Obtain the prediction $O_i$ by (5) given the remaining subgraph.

**6 end**

**7** Run a linear regression model on the generated data $\{Z_i^l, Z_i^r, O_i\}_{i \in [N]}$.

**8** Calculate the importance score for each edge by (6).

---

two directions (denoted as "$l$" and "$r$") of each edge separately; that is, for each undirected edge, we have two independent variables. The values of $z_j^l$ and $z_j^r$ are both binary, where 0 and 1 indicate the 'absence' and 'presence' of the corresponding edge, respectively. Second, we pass the perturbed graph (in specific, the perturbed adjacency matrix) to the fitted GNN, and obtain the corresponding predicted outcome $O$ (for simplicity, it is assumed to be a continuous scalar). Third, we repeat the above sampling procedures for $N$ times, and the generated data are denoted as $\{Z_i^l, Z_i^r, O_i\}_{i \in [N]}$. Finally, based on the newly generated data, any easily interpretable surrogate models can be employed to measure the edge attributions.

In this paper, we choose linear regression as the surrogate model, and the predicted output of node $v$ is decomposed into the linear combination of its neighboring edges,

$$O_i = \alpha + \sum_{j=1}^{m} \beta_j^l z_{ij}^l + \sum_{j=1}^{m} \beta_j^r z_{ij}^r + \epsilon, \tag{5}$$

where $\alpha$ is the intercept, $\beta_j^l$ and $\beta_j^r$ denote the coefficients of the $j$-th neighboring edge in two directions and $\epsilon$ is the error. The intercept term contains two components, i.e., the effects of pure node features and the remaining bias of edges. Model (5) is a standard linear regression problem that can be estimated by the least-squares method. In practice, the predictors may be correlated and the $\ell_2$-regularized estimator is more preferable. The importance score $S_j$ for the $j$-th undirected edge can be simply obtained by averaging the edge coefficients in the two directions,

$$S_j = \frac{1}{2}(\beta_j^l + \beta_j^r). \tag{6}$$

For multi-output tasks, the importance can be taken to be the average of the coefficients with respect to the two directions and multiple outputs. The detail of the proposed edge attribution method is given in Algorithm 1.

**Remark 1** *The formula in (5) concerns the case where the output is a scalar. For multi-output tasks, e.g., multi-classification, the generated data can be viewed as a multi-regression dataset, and the linear regression model in (5) still works.*

The choice of the average number of edges to be removed ($\lambda$) and the number of samples ($N$) are non-trivial for the edge scoring procedure. Removing too many edges each time may lead to very different predictions and it is also difficult to attribute the change of predictions to each removed edge. By default, we set $\lambda = \max\{1, 0.2M\}$.

### 3.3.2   Compact Subgraph Selection

Based on the parsimonious modeling principle, it is preferable to select only a small subgraph containing the target node that can best recover the prediction. Once each edge is properly attributed, the final step is to select a compact subgraph that best approximates the current node prediction.

We rank the $m$ undirected edges in a descending order of the absolute value of their importance scores. Starting with a subgraph only containing the target node $v$, we gradually add the most important edges and corresponding nodes to the subgraph. As the number of edges in the compact subgraph containing the target node $v$ reaches a threshold $T$, the procedure would terminate. The value of $T$ can be determined by the domain knowledge or cross-validation performance of the surrogate linear model.

After the important subgraph is selected, an optional step is to re-estimate the contributions of selected edges. That is, we can repeat the sampling procedure over the selected subgraph, and re-estimate the linear model. To avoid any interpretation difficulty that can be caused by the two directions of each undirected edge, the resampling and re-estimation would treat the two directions of each undirected edge as a whole.

### 3.3.3 Demonstration

After the most important subgraph for the prediction of the target node is identified, the graph visualization can be helpful for interpretation. For instance, positive (or negative) contributions can be colored by deep red (or deep blue); trivial contributions can be painted using transparent gray. The overall benefits of the proposed neighboring edge local interpretation algorithm are summarized below.

(1) We formulate the GNN explanation task as a statistical modeling task; hence, any existing statistical methods can be easily applied.

(2) The contributions of edges can be quantitatively measured, i.e., we can quantify the (positive or negative) contribution of each edge to the final prediction.

(3) Given the edge contribution, the actionable perturbation can be applied to the data. For instance, we may delete one negative edge to increase the current prediction.

For illustration, a demo of the proposed approach is shown in Figure 2. Assume a GNN model is already fitted and we aim to explain how the prediction of node $v_A$ is obtained. We decompose the prediction into (a) the main effects of node $v_A$'s features; and (b) the impact of neighboring edges.

### 3.3.4 The Local Interpretation Extensions

Apart from node classification, the proposed local interpretation can also provide explanations for link prediction with no change in the algorithm. We consider randomly removing both nodes and edges during the sampling process to interpret the graph classification to generate perturbed graph samples. Hence, we could evaluate the node and edge importance for the target graph using the surrogate model.

Moreover, the local interpretation method proposed in this paper is not limited to the GAM-GCN model, and it can be applied to any GNN architecture with complicated decision-making procedures.

## 4 Experiments and Results

The proposed method is examined in numerical studies via four synthetic node classification tasks and four real-world graph classification tasks.

### 4.1 Datasets

We follow the data generation mechanism introduced in Ying et al. (2019), and the following four node classification datasets are used for testing purposes.

(1) Syn1 (BA-Shapes): a multi-classification task consists of a base Barabási-Albert (BA) graph with 300 nodes (class 0) and 80 five-node "house"-structured network motifs (classes 1, 2, and 3 correspond to the top, middle and bottom nodes of the house). The motifs are randomly attached to the base graph. To increase the difficulty, 10% of random edges are added to the final graph. Each node has 10 features which all take the values of 1.

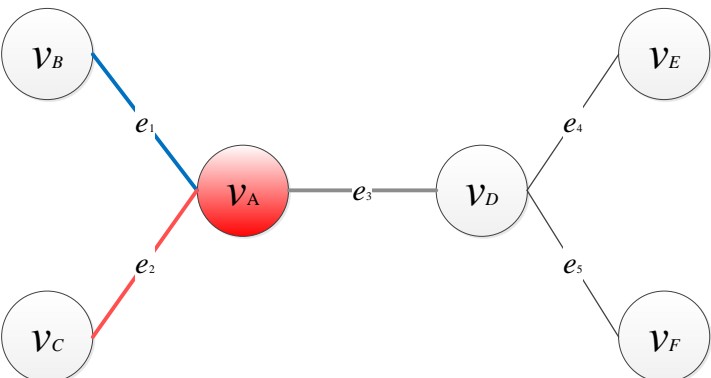

Figure 2: An example graph showing that the main effect of node $v_A$ contributes the most (positive) to the final prediction, followed by edge $e_2$. Edge $e_1$ has a negative contribution (but the magnitude of the absolute value characterizes the importance) to the prediction of node $v_A$. In contrast, edges $e_3, e_4, e_5$ appear to be trivial (or contain no information) for the prediction and hence can be removed from the subgraph.

Table 1: The statistics and properties of the graph classification datasets. The averaged number of nodes per graph, number of node features, number of graphs and number of classes are reported.

|  | Datasets | | | |
|---|---|---|---|---|
|  | Mutag | Graph-SST2 | Graph-SST5 | Graph-Twitter |
| # of nodes | 17.9 | 10.2 | 19.8 | 21.1 |
| # of features | 7 | 768 | 768 | 768 |
| # of Graphs | 188 | 70042 | 11855 | 6940 |
| # of Classes | 2 | 2 | 5 | 3 |

(2) Syn2 (BA-Community): a multi-classification task is simply created by combining two BA-Shapes graphs. Each BA-Shapes graph contains four classes, and the BA-Community data has a total of 8 classes. The standard normal distribution generates the features of each node for each community.

(3) Syn3 (Tree-Cycles): a binary classification task consists of a base 8-level balanced binary tree and 80 six-node cycle motifs. The motifs are randomly attached to the base graph.

(4) Syn4 (Tree-Grid): a binary classification task consists of a base 8-level balanced binary tree and 80 3-by-3 grid motifs. The motifs are randomly attached to the base graph.

Meanwhile, we employ three human-understandable sentiments, one molecular and one vision-based graph datasets to evaluate our method on graph classification datasets. The datasets are introduced below. The statistics and properties of the graph classification datasets are reported in Table 1.

Text sentiment analysis datasets are suitable for graph explanation tasks because people can easily identify the emotional opinion within the text. Therefore, we use the three sentiment graph datasets, Graph-SST5, Graph-SST2, and Graph-Twitter, to explain graph classification tasks (Yuan et al., 2020). The sentiment graph datasets are built based on the text sentiment datasets, the Stanford Sentiment Treebank SST5, SST2, and the Twitter sentiment classification dataset, respectively (Socher et al., 2013; Dong et al., 2014). For each text sequence, each word is regarded as a node and the edges represent the relationships between words, which are extracted by the Biaffine parser. The node embeddings are extracted using a pre-trained BERT model. Each phrase in SST5 is labeled as negative, somewhat negative, neutral, somewhat positive or positive. The labels for SST2 are negative, neutral, or positive. The SST2 dataset is only for binary classification, negative or somewhat negative vs somewhat positive or positive. The Twitter dataset includes the sentiment labels as negative, neutral, or positive.

Mutagenicity is a chemical compound graph classification dataset for drugs, with two classes, mutagen, and non-mutagen. Each graph in Mutagenicity refers to a molecule. The nodes and edges are the atoms and chemical bonds respectively (Morris et al., 2020).

Further, we include one real-world dataset for evaluating the expressive power of GAM-GCN. MNIST-75sp is a graph classification dataset where each original MNIST image are converted to graphs using super-pixels. Super-pixels represent small regions of homogeneous intensity in images Knyazev et al. (2019). It contains 70,000 graphs with 75 nodes each.

### 4.2 Base GNNs

The following basic GNN methods are considered, i.e., the graph convolutional network (GCN), the graph isomorphism network (GIN) (Xu et al., 2018), gated graph convolutional networks (GatedGCN) (Bresson & Laurent, 2017), and graph attention networks (GAT) (Veličković et al., 2018). These methods are all implemented in Python, based on the *torch_geometric* package. All the hyperparameters are configured according to the suggested setting in the corresponding method.

### 4.3 Global Node Feature Interpretation Demonstration

We construct the synthetic node regression dataset as follows. For graph topology, we use the Barabási-Albert Graph on 2000 nodes. The explanatory node feature variables are generated within the domain $[0, 1]^{10}$ from the uniform distribution $U(0, 1)$. For node labels, a nonlinear synthetic function is used to demonstrate the proposed methods as follows,

$$y = 20 \left( x_1 - \frac{1}{2} \right)^2 + 0.15 e^{(-8x_2+4)} + 2 \sin\left(2\pi x_3\right) + 3 \cos\left(4\pi x_4\right) + \varepsilon, \tag{7}$$

where the noise term $\varepsilon \sim \mathcal{N}(0, 0.1)$. In addition to $(x_1, \ldots x_4)$ in the model, $(x_5, \ldots x_{10})$ are noisy variables that have no contribution to the node labels. The ground truth for the feature contributions of $(x_1, \ldots, x_5)$ is shown in Figure 3 (a). By default, the dataset is randomly split into the training (70%), validation (10%), and test (20%) sets. For the specific GAM-GCN architecture, each feature network is equipped with three hidden layers, and the activation function is specified as ReLU. The weight parameters are initialized by the Kaiming Initialization (He et al., 2015). The initial learning rate of the Adam optimizer is set to 0.001. The number of training epochs is set as 500. The learned GAM-GCN is shown to be as accurate as the black-box GNN models on this dataset.

The global overall importance learned by the GAM-GCN is illustrated in Figure 4. The major contributions are from features $x_1, x_2, x_3$ and $x_4$. The ground truth relationship and shape functions learned by an ensemble of GAM-GCN models between individual node features and the response are visualized in Figure 3. Like $x_5$, features $(x_6, \ldots, x_{10})$ do not contribute to the response, which has been confirmed by the GAM-GCN consistently. Hence, we omit them in Figure 3. We observe that the global interpretation of node features by the learned GAM-GCN can successfully capture the relationships in the ground truth.

### 4.4 Expressive Power of GAM-GCN

In addition to the intrinsic interpretability, GAM-GCN preserves the expressive power of the black-box models. The performances of the GAM-GCN model are comparable to the GCN models over the four synthetic datasets. To evaluate the expressivity of GAM-GCN, we compare the performances of GAM-GCN to the 3-layer GCN models on the four synthetic node classification task datasets. For each dataset, the feature networks of GAM-GCN contain three hidden GCN layers with the ReLU activation. The GAM-GCN and GCN are trained with the Adam optimizer for 1000 epochs with a learning rate of 0.001. To measure the performance, we compare the precision, recall, and accuracy over ten runs with different random seeds and simulated datasets, as summarized in Table 2. It is observed that the predictive power of the proposed GAM-GCN is similar to that of the GCN model.

Further, we show that the performances of the GAM-GCN are comparable to the GCN and GAT models over the real-world datasets as summarized in Table 3. For Mutag datasets, by only modeling individual feature

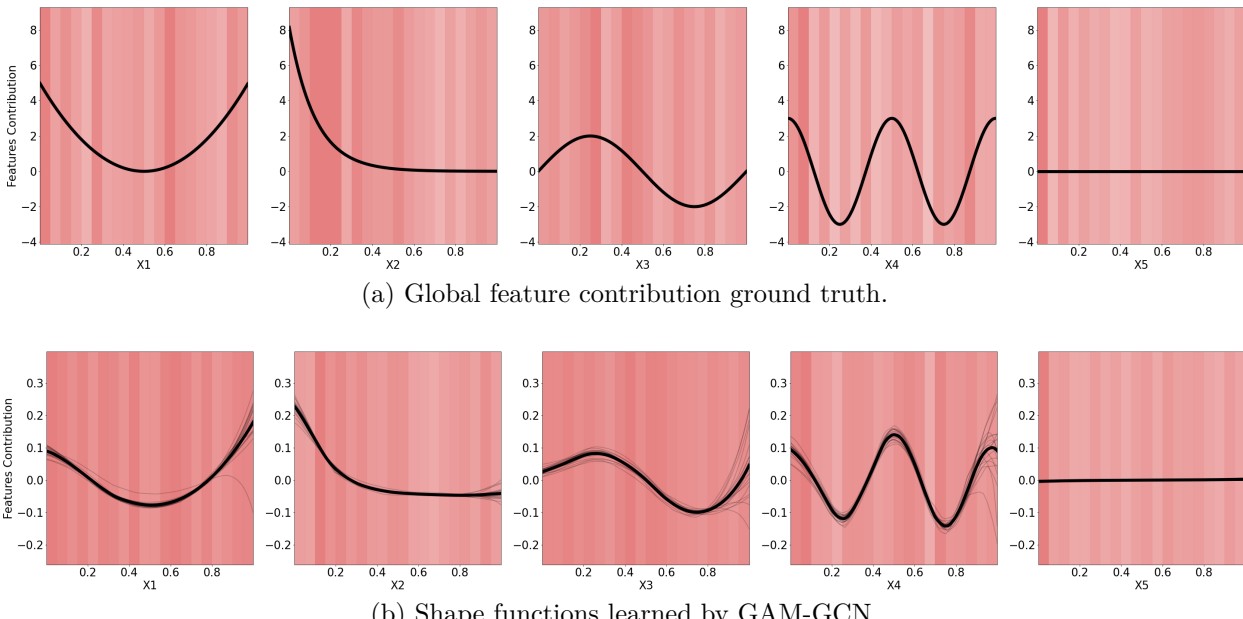

(a) Global feature contribution ground truth.

(b) Shape functions learned by GAM-GCN.

Figure 3: The ground truth and global interpretation of node features by an ensemble of GAM-GCNs on the synthetic node regression dataset. The thin black lines represent different learned shape functions from the ensemble to show the consistency of the members of the ensemble. The data density for each feature is visualized using the pink bars: The darker the bar, the more data with that value.

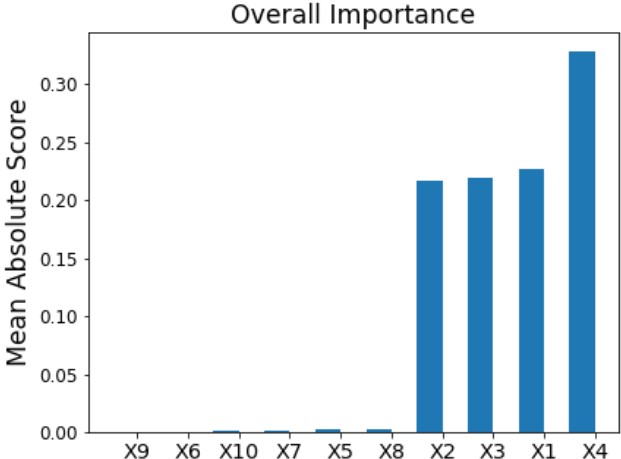

Figure 4: The overall importance learned on the synthetic node regression dataset.

effects GAM-GCN achieves comparable performances. However, we observe a significant increase in accuracy as we further consider the pairwise interactions between node features using GAM-GCN for MNIST-75sp because the interactions of coordinates and pixel values are essential for learning visual concepts.

Table 2: The performance of our GAM-GCN model on the synthetic node classification datasets. Below are the average and standard deviation (in brackets) of the precision, recall, and accuracy (%) on the test sets over ten runs using different random seeds with the simulated datasets for the GCN and GAM-GCN models.

| Datasets | Metrics | **GCN** | **GAM-GCN** |
|---|---|---|---|
| Syn1 | precision | 96.1 (2.9) | 95.9 (3.3) |
| | recall | 94.2 (4.1) | 94.1 (4.6) |
| BA-Shapes | accuracy | 95.6 (2.4) | 96.5 (2.6) |
| Syn2 | precision | 66.9 (7.3) | 67.92 (4.3) |
| | recall | 64.8 (3.3) | 67.75 (4.6) |
| BA-Community | accuracy | 77.9 (3.7) | 74.88 (3.1) |
| Syn3 | precision | 89.2 (5.2) | 97.8 (1.5) |
| | recall | 89.7 (5.7) | 95.4 (3.6) |
| Tree-Cycles | accuracy | 89.6 (5.2) | 97.1 (2.2) |
| Syn4 | precision | 87.4 (2.5) | 87.4 (5.8) |
| | recall | 86.3 (2.2) | 87.4 (7.0) |
| Tree-Grid | accuracy | 87.4 (2.0) | 88.1 (5.6) |

## 4.5 Experiments on Edge Attribution

### 4.5.1 Edge Attribution Evaluation Metrics and Benchmarks

For edge attribution, the following three measurements are considered. (1) AUC: The area under the ROC curve score of $m$ edge attributions compared to the ground truth; (2) Accuracy: we compare the accuracy of edges in the selected compact subgraph as compared to the ground truth graph structure; (3) Fidelity: The approximation error between the prediction under the selected compact subgraph and the original prediction given the full graph, which is defined as

$$\text{Fidelity} = \frac{1}{N} \sum_{i=1}^{N} \left| g\left(\mathcal{G}_i\right)_{y_i} - g\left(\mathcal{G}_i^{\text{sub}}\right)_{y_i} \right|, \tag{8}$$

where $g$ represents the GNN model to explain, $\mathcal{G}_i$ and $\mathcal{G}_i^{\text{sub}}$ denotes the sample $i$ with original graph and selected compact subgraph, $y_i$ stands for the ground truth label.

We compare with the following baseline methods, the gradient-based edge attribution, GNNExplainer (Ying et al., 2019), PGExplainer (Luo et al., 2020), and DeepLIFT (Shrikumar et al., 2017). These methods are all implemented in Python, based on the *torhc_geometric* package. All the hyperparameters are configured according to the suggested setting, and for a fair comparison, each method is limited to the interpretation of the $K$-hop neighboring subgraph.

Table 3: The prediction performance of GAM-GCN on real-world datasets. Below is the average and standard deviation (in brackets) of accuracy (%) on the test sets over ten runs with different random seeds for the GCN, GAT, and our intrinsically interpretable model without or with the second-order node feature interactions.

| Models | Mutag | MNIST-75sp |
|---|---|---|
| GCN | 73.3 (5.8) | 90.1 (0.2) |
| GAT | 75.0 (5.0) | 95.5 (0.2) |
| GAM-GCN | 76.5 (3.7) | 36.8 (0.5) |
| GAM-GCN-I | 81.4 (2.9) | 92.7 (0.1) |

### 4.5.2 Node Classification Model Explanation Results

To assess the performance of our models for local interpretation of neighboring edges and edge attributions, benchmarks on test sets are compared according to AUC, accuracy, and fidelity, as summarized in Table 4. Because the motifs for explaining the four synthetic datasets are known, we can set the size of the selected subgraph to be that of the ground truth. It is observed that our proposed local interpretation method for neighboring edges generally outperforms the benchmarks.

We further visualize the instance-level explanation results for target nodes in the Syn1 dataset correctly predicted by the GCN model in Figure 11. Additionally, the instance level graph structure explanations for target node 475 of the Syn2 dataset are illustrated in Figures 6, where we observe that the GCN model learned some hidden intrinsic properties unexpectedly, that the middle part of the house is always connected with the base graph.

Furthermore, the interpretation results are shown in Figure 12 for the target nodes incorrectly predicted by the GCN model, where our proposed local interpretation algorithm highlights the partial substructure of the predicted class and implies the decision-making process of the GCN model for the target node. More visualization results can be found in the supplementary material.

Table 4: The performance of our local interpretation for neighboring edges and benchmarks are compared according to AUC, accuracy, and fidelity.

| Datasets | Metrics | **Ours** | **GNNExplainer** | **PGExplainer** | **Grad** | **DeepLift** |
|---|---|---|---|---|---|---|
| Syn1 | AUC | **0.9905** | 0.8273 | 0.9619 | 0.9781 | 0.2665 |
| | top-$k$ accuracy | 0.8938 | 0.7850 | **0.9087** | 0.8985 | 0.4607 |
| (BA-Shapes) | top-$k$ fidelity | **0.0964** | 0.1675 | 0.1617 | 0.1167 | 0.4596 |
| Syn2 | AUC | **0.9678** | 0.8699 | 0.9191 | 0.9625 | 0.5572 |
| | top-$k$ accuracy | **0.7758** | 0.6306 | 0.7107 | 0.7344 | 0.5888 |
| (BA-Community) | top-$k$ fidelity | **0.1219** | 0.1677 | 0.1964 | 0.1724 | 0.2468 |
| Syn3 | AUC | **0.8864** | 0.6444 | 0.2255 | 0.8009 | 0.2676 |
| | top-$k$ accuracy | **0.8745** | 0.7499 | 0.6039 | 0.8242 | 0.6314 |
| (Tree-Cycles) | top-$k$ fidelity | 0.4633 | 0.5412 | 0.4784 | 0.5640 | **0.4592** |
| Syn4 | AUC | 0.8448 | 0.7172 | 0.5347 | **0.8737** | 0.4691 |
| | top-$k$ accuracy | 0.7689 | 0.7086 | 0.5997 | **0.7825** | 0.5961 |
| (Tree-Grid) | top-$k$ fidelity | **0.3212** | 0.5091 | 0.4184 | 0.4044 | 0.8223 |

### 4.5.3 Graph Classification Models Explanation Results

The proposed edge attribution method can be easily applied to graph classification tasks. We use the 3-layer GCN model as the graph classifier. The interpretation results for a text sequence in Graph-SST2 are visualized in Figure 8. The semantic of the text, a brief movie review, is labeled as positive and predicted correctly by the classifier. On the contrary, the movie storyline about the marriage issue is described using negative terms: '*silly*' and '*crude*'. Our model successfully captures the contrast between the semantics of the review and the storyline. Our model identifies that the substructure, '*what I like about*', has the largest positive contributions, while the adjectives for the storyline: '*silly*' and '*crude*', have the most prominent adverse effects. GNNExplainer, PGExplainer and the gradient-based edge attribution methods cannot distinguish such contrast since the calculated importance scores for the substructures are all positive.

Moreover, we compare the fidelity of our method and the benchmarks under different sparsity levels. Sparsity is the proportion of the graph that has been recognized as essential for prediction. A larger fraction of the subgraph is selected to be an explanation when the sparsity is set to be lower. Hence, lower sparsity results in higher fidelity using the same explainable GNN model. The quantitative results for explaining the 3-layer GCN graph classifier are summarized in Figure 9. Our proposed method outperforms the benchmarks for all four datasets under various sparsity levels. Further, we conduct similar experiments using a 3-layer graph isomorphism network (GIN, Xu et al., 2018) as a graph classifier. The gradient-based edge attribution cannot be applied to explaining GIN since the neighborhood aggregation strategy of GIN does not utilize the edge

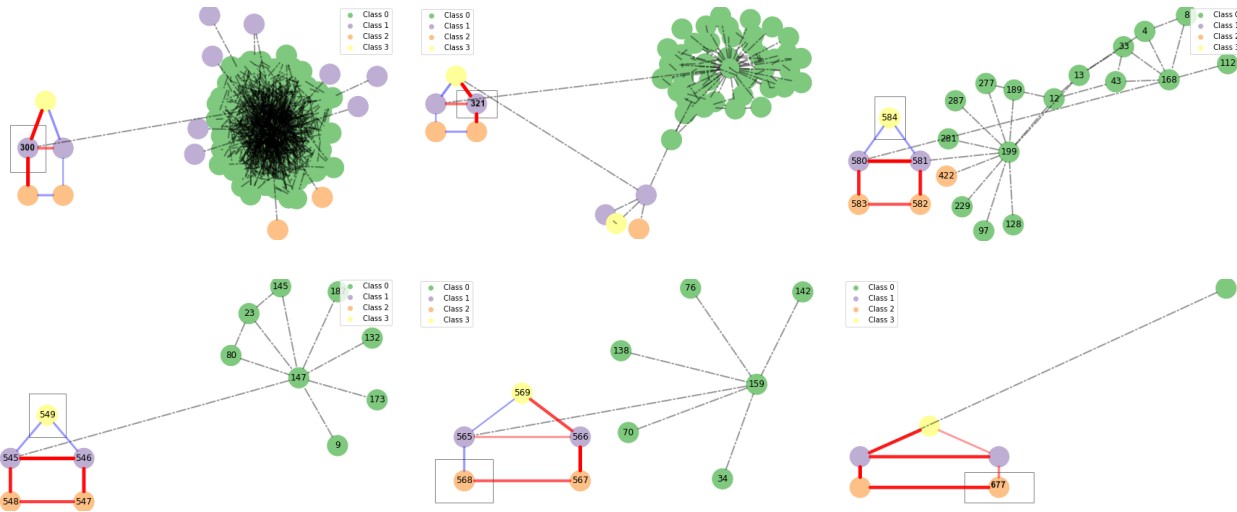

Figure 5: Explanation results of the Syn1 (BA-Shapes) dataset for the target nodes correctly predicted by the GCN model. A box encloses the target node. The edges with the most significant positive and negative contributions are marked in red and blue, respectively.

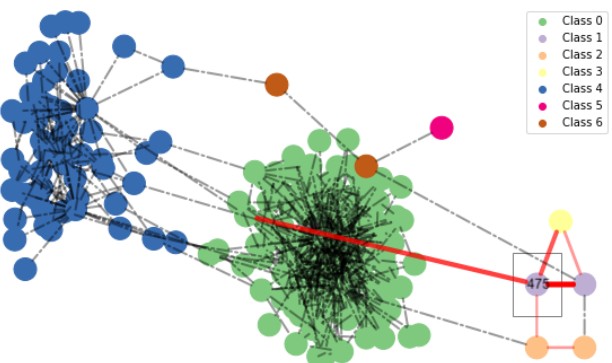

Figure 6: Instance level graph structure explanation for node classification for the 3-hop neighboring subgraph of node 475 in the Syn2 dataset.

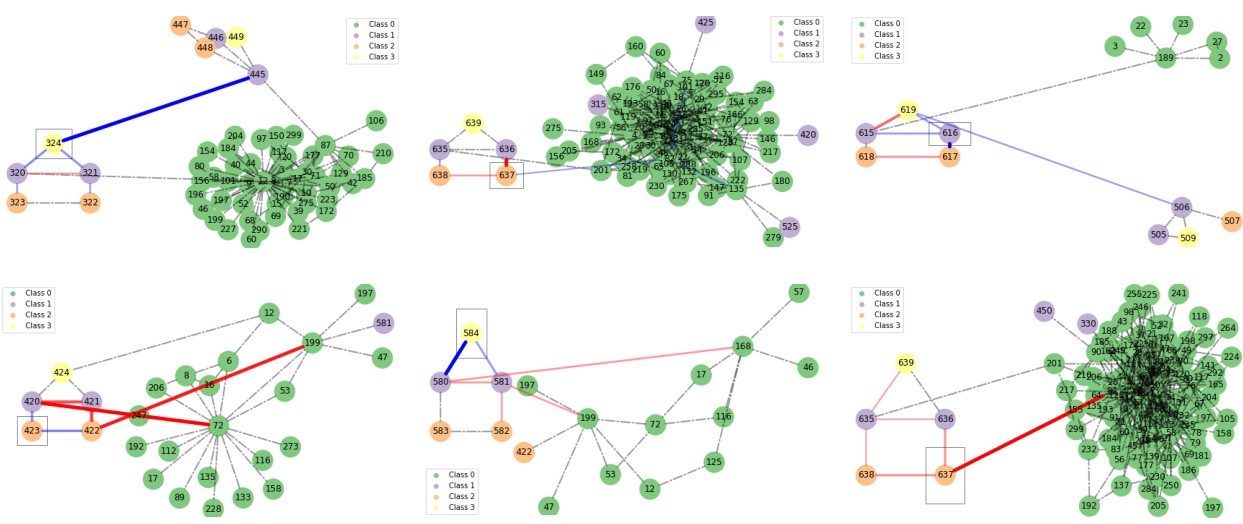

Figure 7: Explanation results of the Syn1 (BA-Shapes) dataset for the target nodes incorrectly predicted by the GCN model. A box encloses the target node. The edges with the most significant positive and negative contributions are marked in red and blue, respectively.

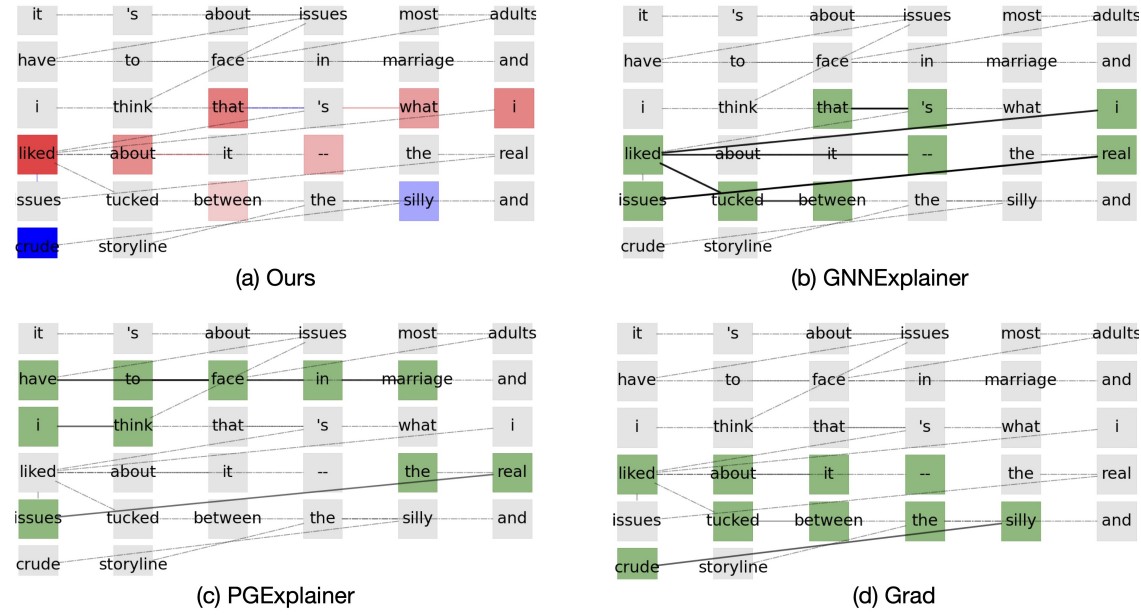

Figure 8: Interpretation results on the Graph-SST2 dataset with a GCN graph classifier. The text sequence is "*it's about issues most adults have to face in marriage and i think that's what i liked about it – the real issues tucked between the silly and crude storyline*". The semantic of the sample, a brief movie review, is labeled and predicted as positive. However, the movie storyline about the marriage issue is described using negative terms: *silly* and *crude*. The important positive and negative contributions are respectively indicated in red and blue using our method. The other explanation methods cannot indicate the contributions' directions; thus, the important substructures are marked in green.

weight of a graph. We visualize the quantitative results for explaining GIN in Figure 10. Our interpretation method consistently achieves lower fidelity across various sparsity.

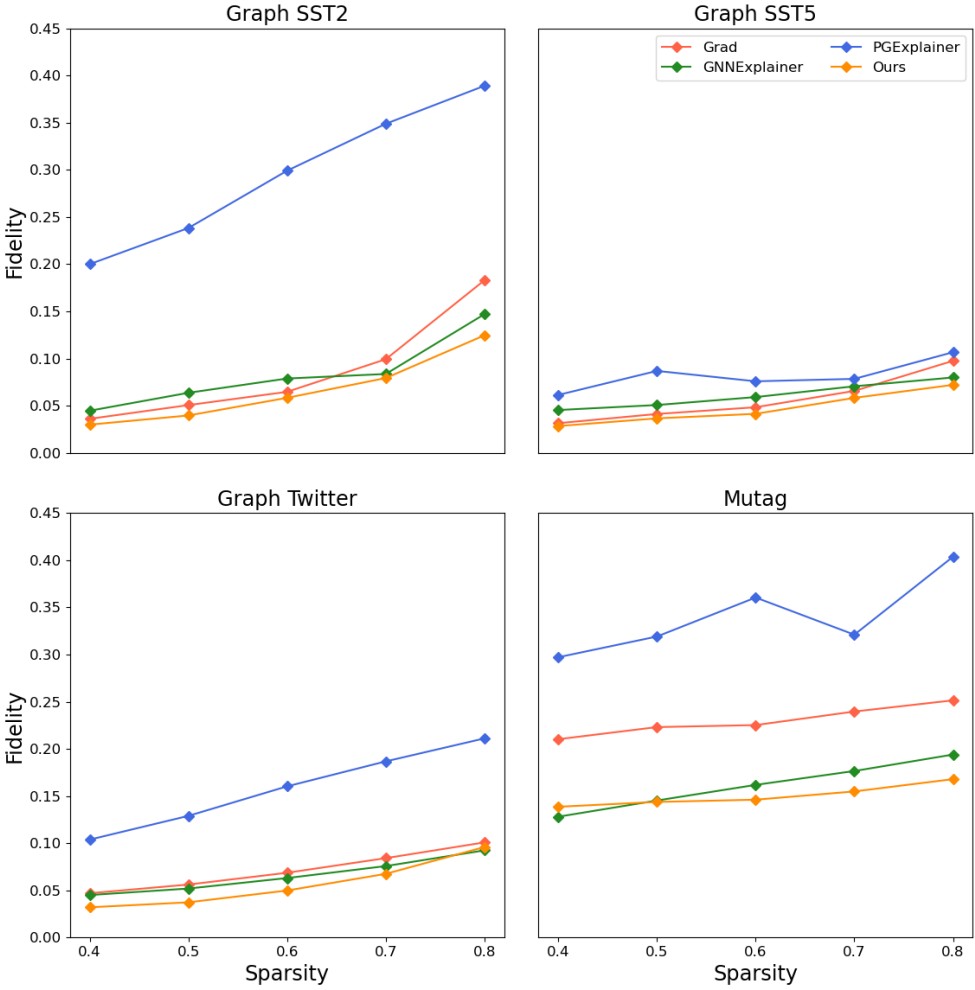

Figure 9: The quantitative studies for different GNN interpretation methods for graph classification. We evaluate the fidelity of the explanation for a 3-layer GCN model on the four graph classification tasks under different sparsity levels. Fidelity is defined as the approximation error between the prediction under the selected compact subgraph and the original prediction by the full graph.

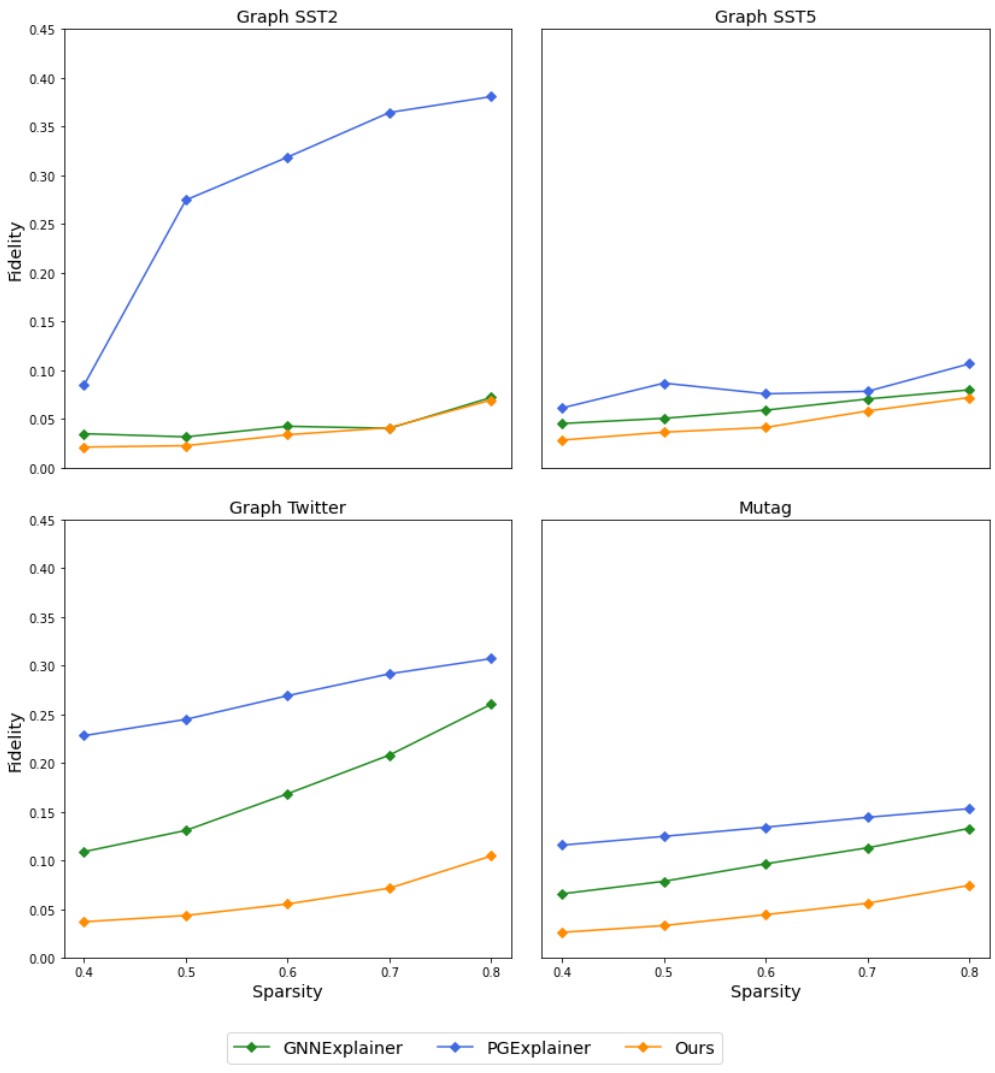

Figure 10: The quantitative studies for different GNN interpretation methods for graph classification. The 3-layer GIN is used as the base model for graph classification. We evaluate the fidelity of the four graph classification tasks under different sparsity levels.

Table 5: Computational time comparison for edge attribution methods. Below are the average and standard deviation of the Computation time for producing local interpretation for each node in Syn1 (BA-Shapes) in seconds over ten runs.

|  | **Ours** | **GNNExplainer** | **PGExplainer** |
|---|---|---|---|
| GCN | 208 (41) | 260 (55) | 425 (44) |

### 4.5.4 Efficiency Analysis for Edge Attribution

We evaluate the efficiency under the same computational resource by comparing the computational time to perform local interpretation for all nodes within Syn1 (BA-Shapes) dataset. Our edge attribution method is slightly more efficient compared with the learning-based methods GNNExplainer and PGExplainer and Gradient-based methods as illustrated in Table 5.

## 5 Conclusion

This paper proposes a new tool for interpreting graph neural networks. The main effect of each node feature is captured by a sub-network containing one input neuron, multiple hidden layers, and one output neuron, which can explain the pure contribution of node features. Moreover, the contributions of neighboring edges are measured by post-hoc surrogate linear models. The importance of edges is then quantified by the regression coefficients, and hence the most important subgraph for the prediction can be selected. Experiments conducted on multiple synthetic and real-world datasets show that the proposed method is highly interpretable with minor sacrifice on prediction performance.

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

# A   Appendix

## A.1   Visualizations of Explaining Node Classification

We present more visualizations of the explanation results for the four synthetic node classification datasets, Syn1 (BA-Shapes), Syn2 (BA-Community), Syn3 (Tree-Cycles), and Syn4 (Tree-Grid). The instance-level explanation results for target nodes correctly predicted by the GCN model are shown in Figures 11, 13, 15 and 17, respectively. For the target nodes with the correct predictions, the proposed local interpretation for neighboring edges can generally capture the corresponding motifs of the datasets. Meanwhile, the GCN models may have used some unexpected shortcuts during prediction. For example, certain nodes in the motifs are always connected with the base graphs. Our proposed local interpretation algorithm for neighboring edges can highlight those shortcuts learned by the GCN model.

For the target nodes incorrectly predicted by the GCN model, the interpretation results are shown in Figures 12, 14, 16 and 18. Our proposed local interpretation algorithm highlights the partial substructure of the predicted class, which would imply the decision-making process of the GCN model for the target node.

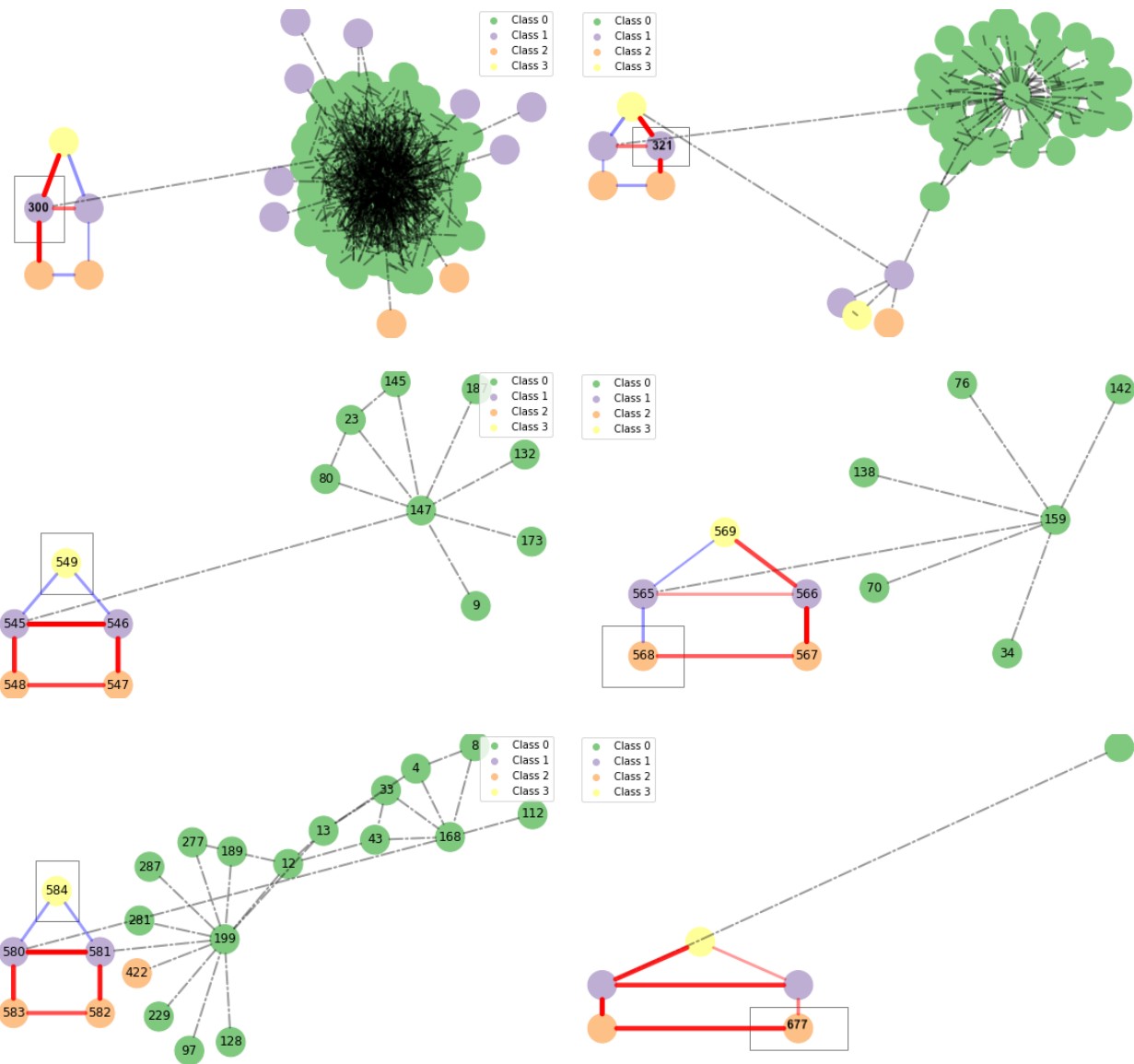

Figure 11: Explanation results of Syn1 (BA-Shapes) dataset for the target nodes **correctly** predicted by the GCN model. The target node is enclosed by a box. The edges with the most significant positive and negative contributions are marked as red and blue, respectively.

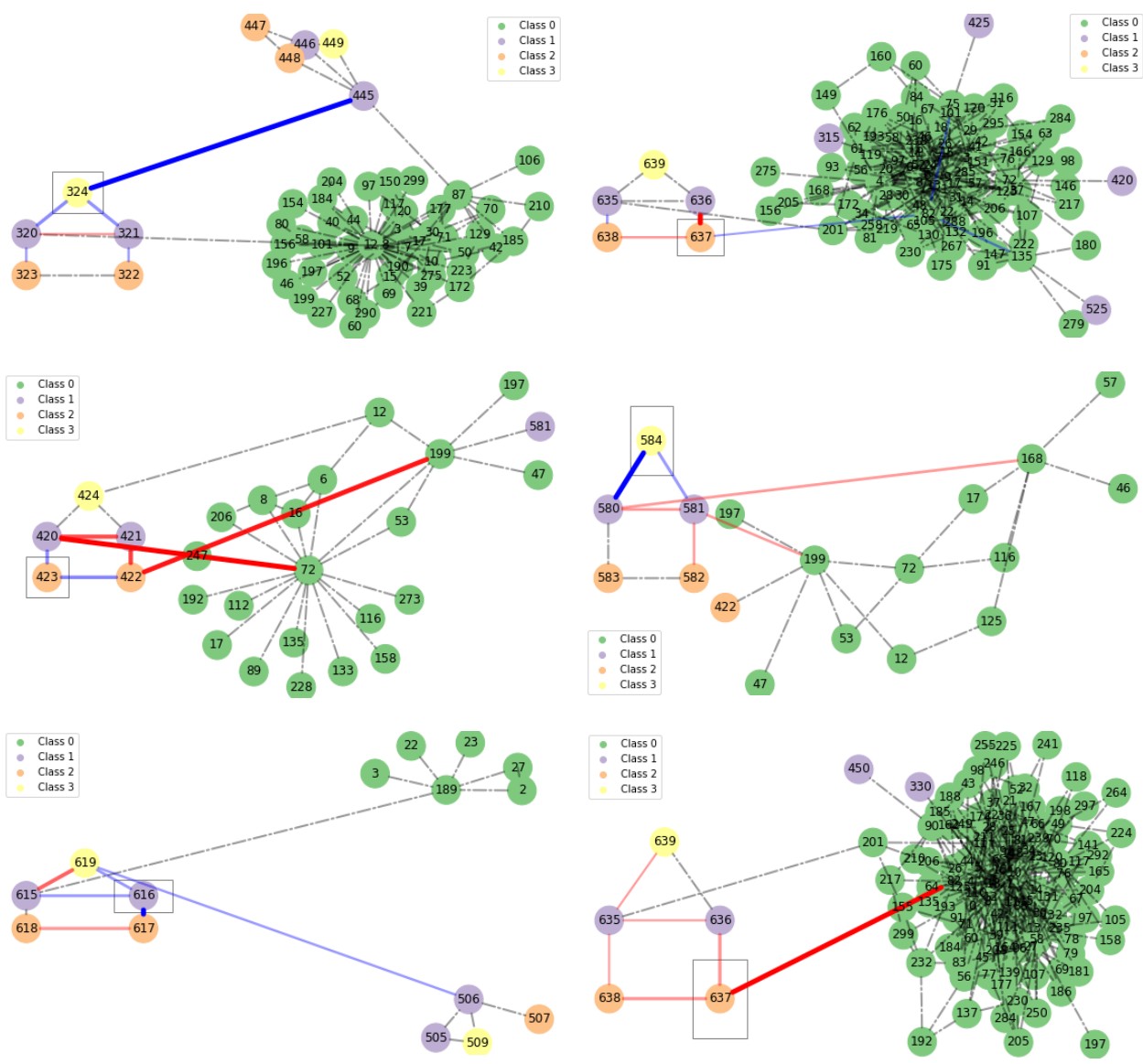

Figure 12: Explanation results of Syn1 (BA-Shapes) dataset for the target nodes **incorrectly** predicted by the GCN model. The target node is enclosed by a box. The edges with the most significant positive and negative contributions are marked as red and blue, respectively.

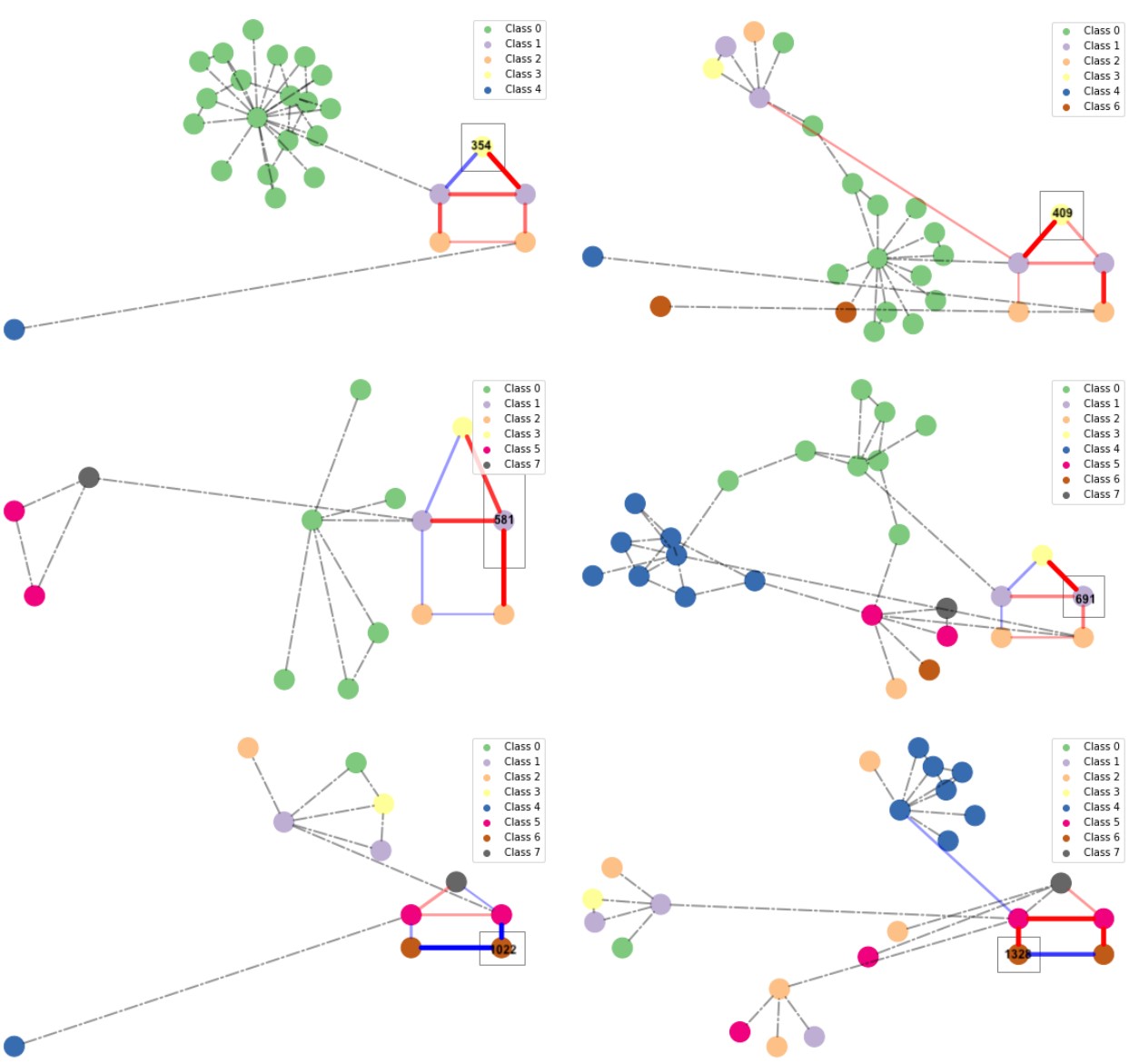

Figure 13: Explanation results of Syn2 (BA-Community) dataset. The target nodes are **correctly** predicted by the GCN model. The target node is enclosed by a box. The edges with the most significant positive and negative contributions are marked as red and blue, respectively.

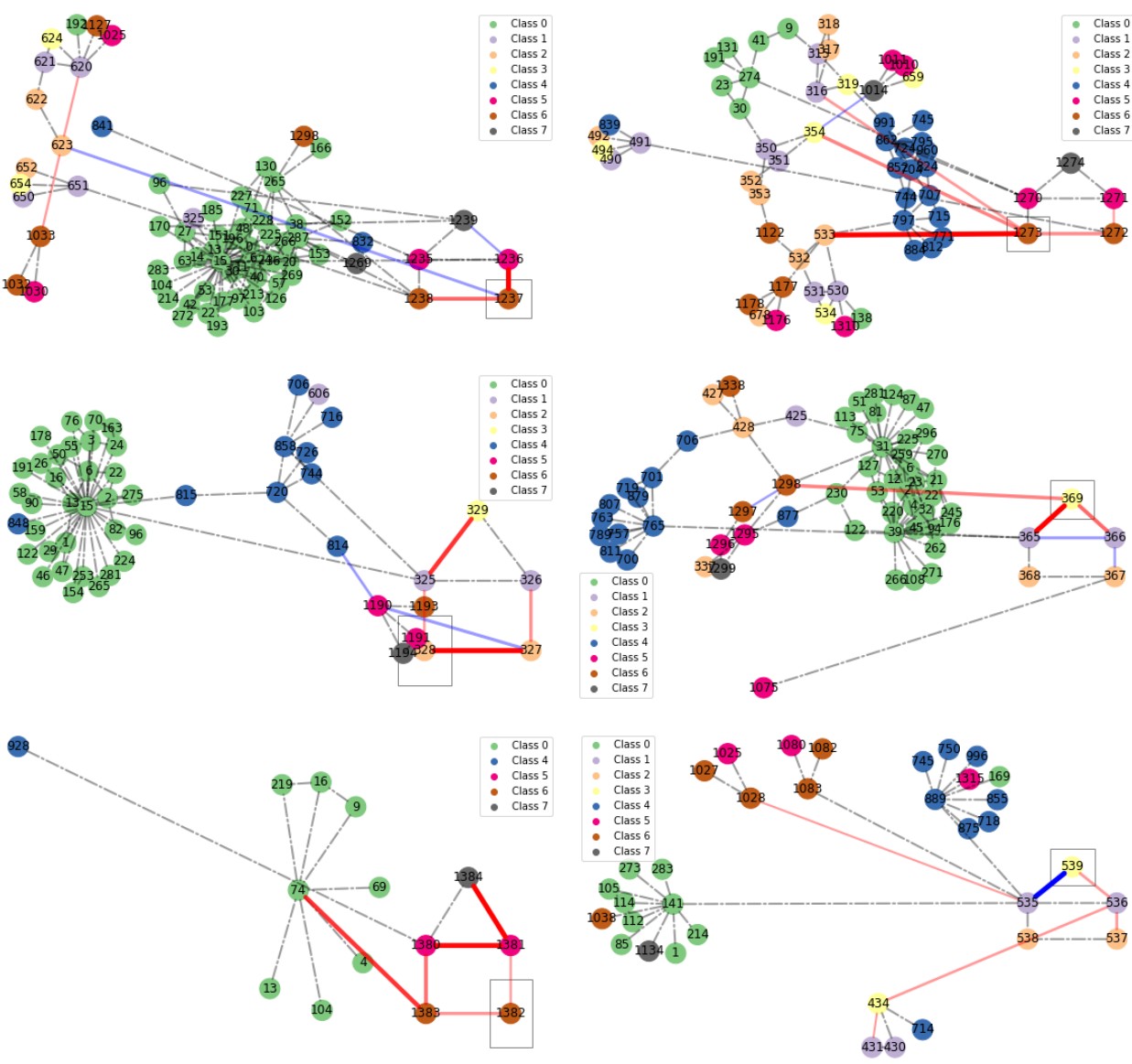

Figure 14: Explanation results of Syn2 (BA-Community) dataset for the target nodes **incorrectly** predicted by the GCN model. The target node is enclosed by a box. The edges with the most significant positive and negative contributions are marked as red and blue, respectively.

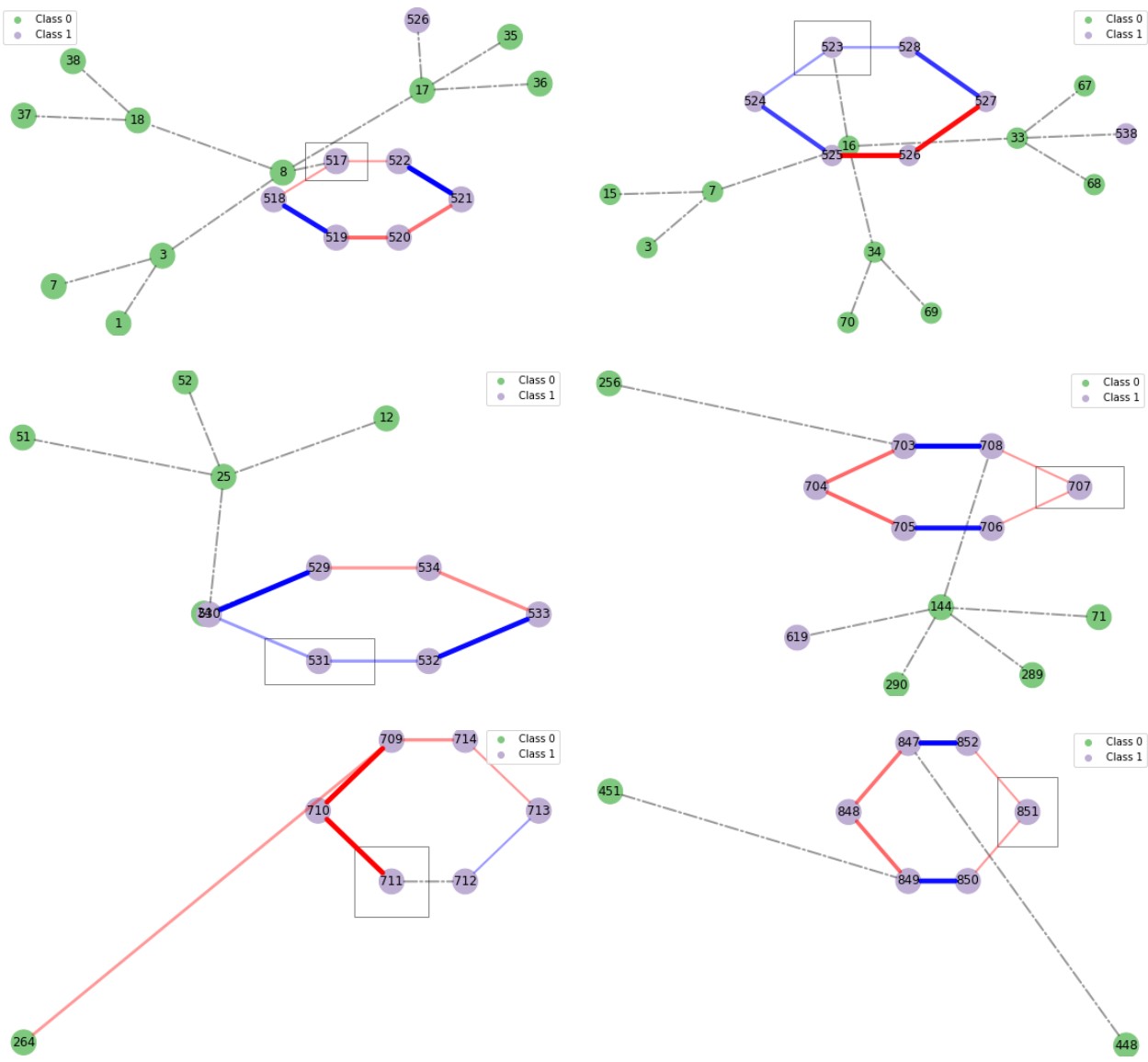

Figure 15: Explanation results of Syn3 (Tree-Cycles) dataset for the target nodes **correctly** predicted by the GCN model. The target node is enclosed by a box. The edges with the most significant positive and negative contributions are marked as red and blue, respectively

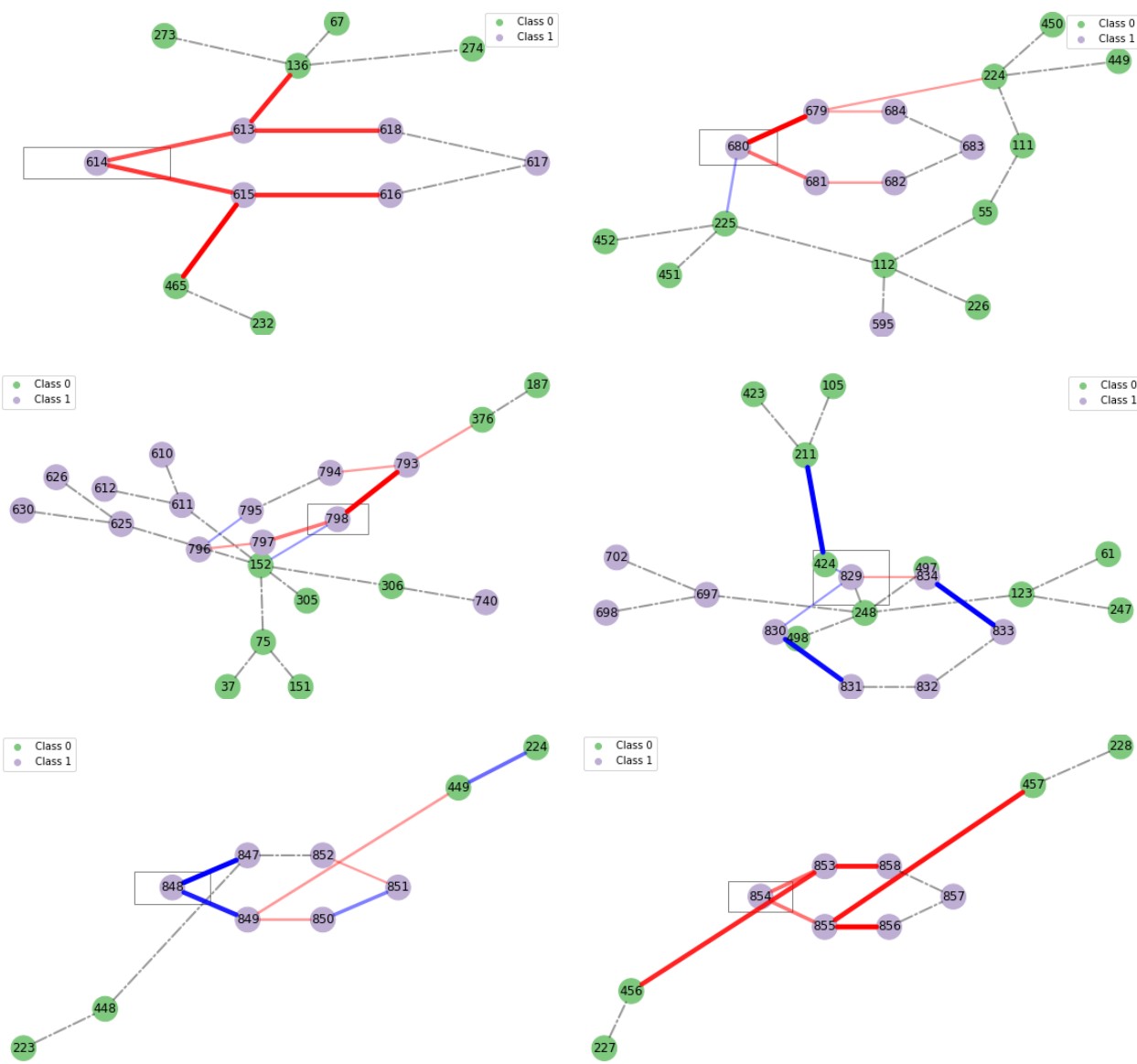

Figure 16: Explanation results of Syn3 (Tree-Cycles) dataset for the target nodes **incorrectly** predicted by the GCN model. The target node is enclosed by a box. The edges with the most significant positive and negative contributions are marked as red and blue, respectively

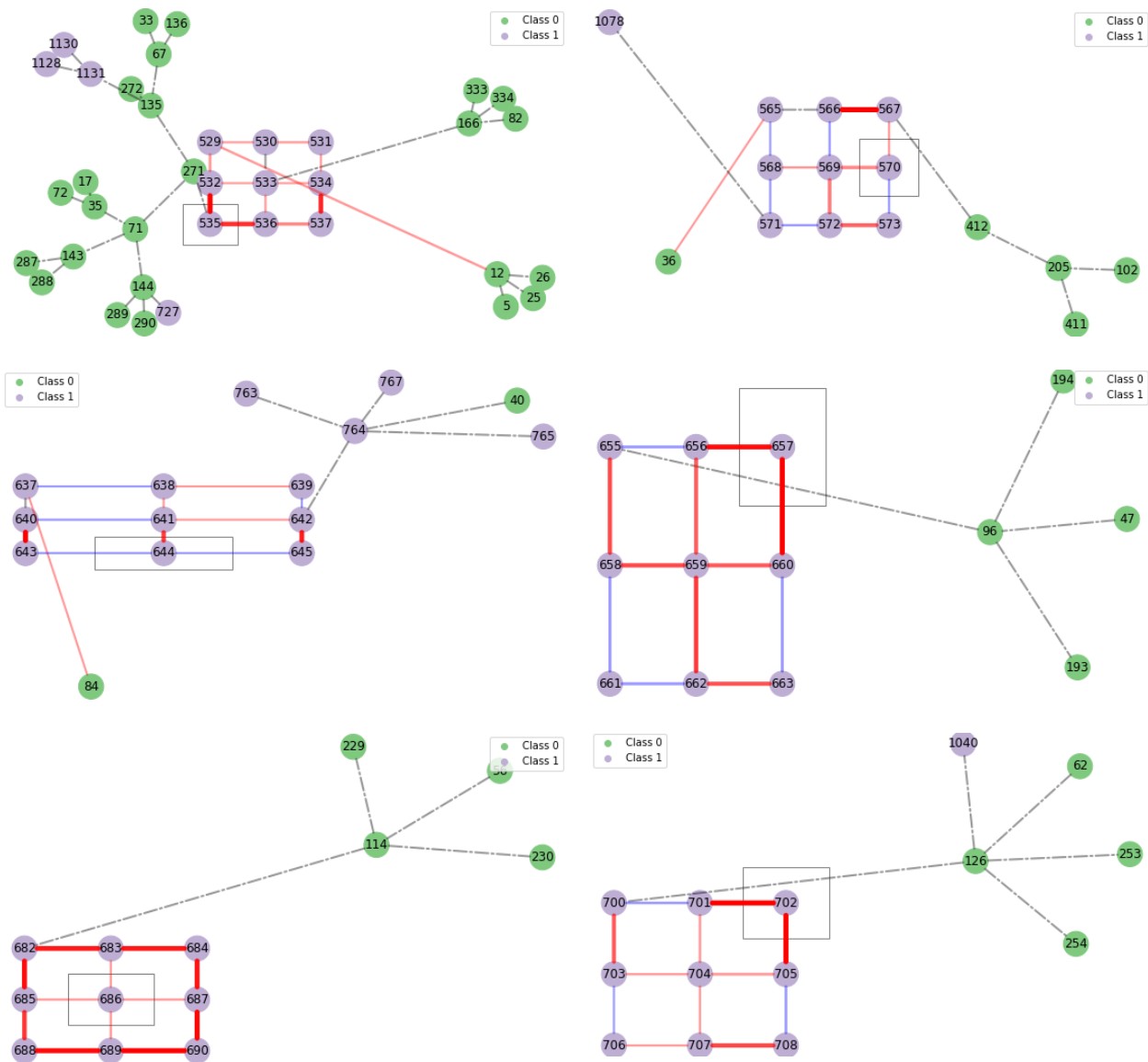

Figure 17: Explanation results of Syn4 (Tree-Grid) dataset for the target nodes **correctly** predicted by the GCN model. The target node is enclosed by a box. The edges with the most significant positive and negative contributions are marked as red and blue, respectively

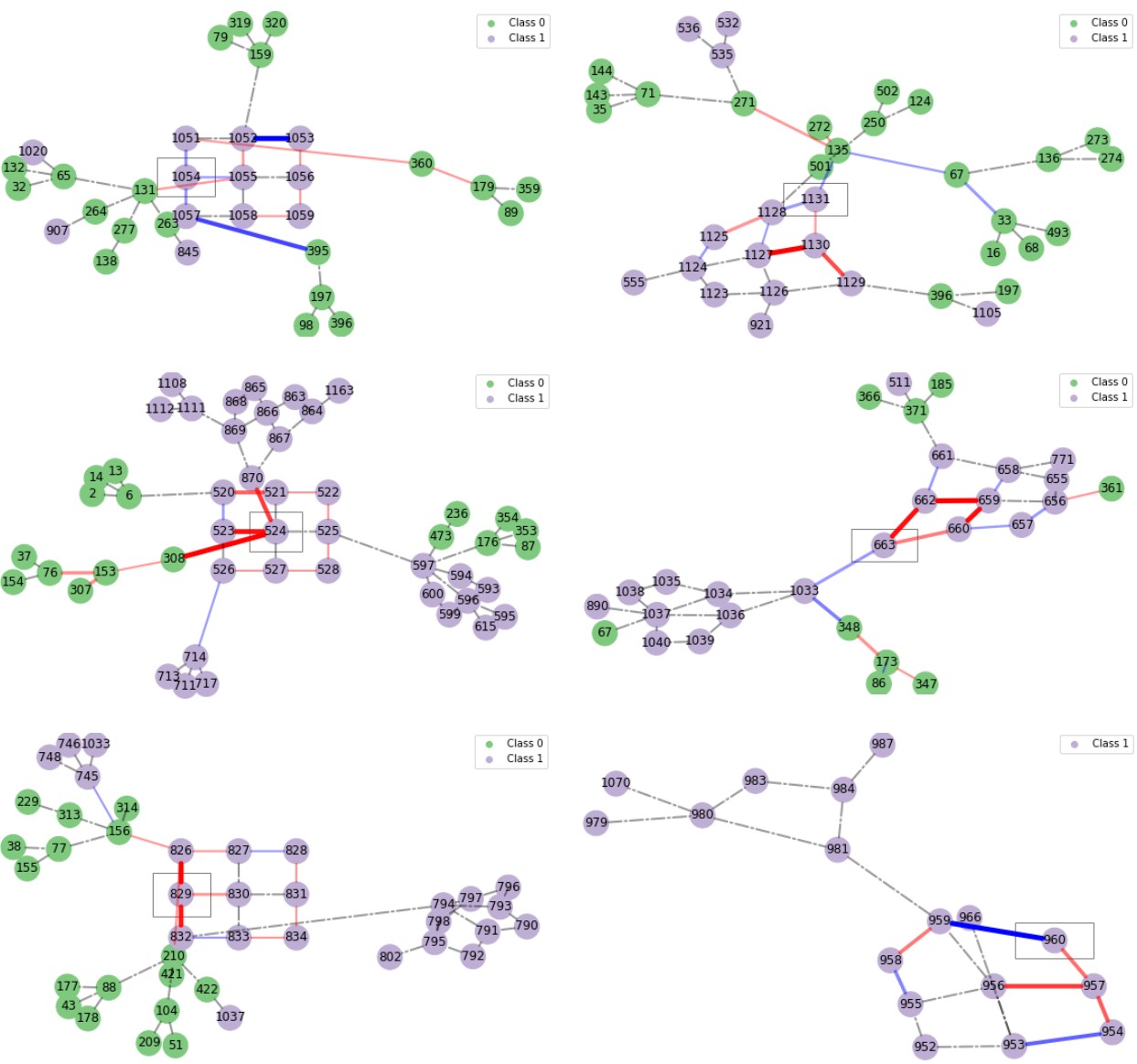

Figure 18: Explanation results of Syn4 (Tree-Grid) dataset for the target nodes **incorrectly** predicted by the GCN model. The target node is enclosed by a box. The edges with the most significant positive and negative contributions are marked as red and blue, respectively

