# OpenReview forum: "Unlock the Black Box by Interpreting Graph Convolutional Networks via Additive Decomposition"
_TMLR — Rejected by TMLR_

### Review · Reviewer_7wdS · 2023-05-29

**Summary Of Contributions:**

This paper presents a method to train and interpret graph neural networks. It decomposes the original GNN into a generalized linear model (GAM), which involves multiple GNNs, each for a single feature dimension, to globally interpret the effect of different feature dimensions. It then proposes to locally estimate the effects of edges by perturbing the local graph structure of a node, and to establish the prediction results with the absence/presence of an edge. Experiments are done over synthetic node classification datasets and real-world graph classification datasets, where the proposed GAM-GCN performs better than explainable GNN baselines.

**Audience:**

Yes

**Broader Impact Concerns:**

No.

**Claims And Evidence:**

No

**Requested Changes:**

Please try to clarify and address the "Weaknesses" and "Minor Weaknesses and Questions".

**Strengths And Weaknesses:**

### Strengths:
- The studied topic, explainable graph neural networks, is important in building trustworthy AI and graph mining systems in real life.
- The proposed method seems simple and sound. I specifically find the decomposition of node feature importance and neighbor (edge importance) reasonable, as some graphs are heterophilic where immediate neighbors are not necessarily similar to each other. In this case, edge contributions are low but the node features still contribute a lot.
- Experiments are done on real-world datasets with some quantitative metrics. As explainable AI is a somewhat subjective field, I appreciate it.
- Figures 3 and 4 provide a clear illustration about how the learned GAM-GCN fits with the ground truth (with a synthetic dataset).

### Weaknesses:
1. Missing baselines. I am not very familiar with explainable GNN, but as far as I know, there are other methods that build explainable GNNs with decomposition, such as (Feng et al. 2022). The authors are advised to discuss and compare it if possible.

(Feng et al. 2022) DEGREE: Decomposition based explanation for graph neural networks. ICLR 2022

2. Regarding Eqn. 4 about the explanation of node features, it seems that the authors assume that "node features with higher variance in $f_j$ across nodes are important", as MAE measures how a random variable deviates from its mean value. It makes some sense but I have several questions about it.
    - First, the $f_j^k$ may be poorly learned, such that the $f_j$ are just random noises. In this case, will the explanation fail?
    - Second, what is the rationale of using MAE, instead of other metrics such as RMSE to better show the variance? Any experiments to support that?
    - Finally, what is the rationale to first train $f_j$ with the edges, and then remove all edges to evaluate feature importance? It seems that it is also OK to just train the models to fit the labels with individual features and use no neighbor information, and do explanation using these models. Any justifications about the design?

3. For Eqn. 5 to learn edge attributions, the authors seem to assume that different edges bring additive influence to the center node (as evidenced by the linear regression model in Eqn. 5). I would like to know how the assumption holds in practice, as intuitively, different edges may act non-linearly to impact a node.

4. It may be time consuming to perform Algorithm 1 to all nodes in the graph. I am not an expert in explainable GNN, and would like to see a comparison in terms of efficiency between GAM-GNN with the baselines. Is GAM-GNN less efficient than baselines?

5. It seems that the authors experiment with synthetic datasets only for node classification. It would be better if some results can also be performed with real-world node classification datasets, especially about the expressive power.

### Minor weaknesses and questions:
- It seems that the authors deal with bi-directional edges by splitting them into two directional ones in Eqn. 5 ($Z^l, Z^r$). However, is this compatible with the symmetric normalization in Section 3.1?

- It would be better if the authors can discuss why we need to re-estimate the contributions after selected edges. Will they differ significantly before and after selections? Any results to show that?

- In Eqn. 1, the authors use $g(\mathbb{E}(Y|X)) = \beta_0 + \sum_{j=1}^d h_j(X_j)$, and say that $g$ is a link function. However, it does not seem clear why is $g$ necessary here.

- I am interested in how compact subgraph selection can improve classification accuracy. In general, if some noisy edges are removed, the classification accuracy should increase a little. Is this the case?

---

> ### Author Response · Authors · 2023-07-28
>
> Dear Reviewer 7wdS,
>
> Thank you for your detailed and insightful review of our paper. We appreciate your comments on the strengths of our work and your thoughtful suggestions for improvement. We revised the manuscript to address your concerns and provided explanations and clarifications where needed.
>
> 1.	Missing baselines: We acknowledge the importance of including relevant baselines for comparison. We included the discussion of DEGREE in the related work. DEGREE decomposes the information flow in a trained GNN model to track the contributions of different components of the input graph to the final prediction. It designs specific decomposition schemes for commonly used GNN layers like graph convolution and graph attention layers. In summary, our method leverages more extensive statistical theory for interpretability, while DEGREE relies more on intuitive decomposition rules and properties of GNN propagation. For explaining graph structures on the synthetic node classification datasets, our simple edge attribution method are comparable to the DEGREE method (Feng et al., 2022).
>
> 2.	Regarding Eqn. (4): Your concerns about the assumptions and metrics we used in Eqn. (4) are valid. First, we agree that the explanation is valid only when the GAM-GCN preserves adequate expressive power. Thus, we evaluate the node classification performance of GAM-GCN and black-box GCN models on four synthetic datasets, showing comparable results. Further, for Equation (4), we follow the MAE design and implementation of global importance in the Neural Additive Model (Agarwal et al., 2021).
>
> Agarwal, Rishabh, et al. "Neural additive models: Interpretable machine learning with neural nets." Advances in neural information processing systems 34 (2021): 4699-4711.
>
> 3.	Regarding Eqn. 5: We understand your concerns about the additive assumption for edge attributions. We could consider the effect of the co-occurrence of edges by further involving pairwise interaction terms as well as polynomial terms in Eqn. 5. Meanwhile, we show that only modeling the individual effects has already been able to achieve lower fidelity (predictive difference between the original graph and selected graph) under the same sparsity compared with other methods based on the empirical results in Figs 9 and 10.

---

> > ### Author Response · Authors · 2023-07-28
> >
> > 4.	Efficiency comparison with baselines: Thank you for your suggestion. We included a comparison of the efficiency of our edge attribution method with existing baselines in Table 3 of Section 4.5.4 to provide a more comprehensive evaluation of our method. Based on Table 3, our method is slightly more efficient compared with the base line methods using one V100 GPU, taking 208 seconds to explain all nodes in the motif for SYN1 dataset.
> > 5.	Real-world node/graph classification datasets: We agree that real-world node/graph classification datasets would provide more insight into the expressive power of our method. Actually, we found that with pairwise interactions, our GAM-GCN with intrinsic interpretability could have equivalent performance compared with GNN models. We include additional experiments in Table 3 on real-world graph datasets, MUTAG and MNIST-75sp, in the revised manuscript, where the GAM-GCN with pairwise interactions shows similar performances compared with GCN and GAT.
> > Minor Weaknesses and Questions:
> > 1.	Bi-directional edges and symmetric normalization in section 3: Empirically, we observed that considering the directions when modeling edge attribution results in better performance when fitting the output variation compared with those modeling the undirected edges. Symmetric normalization of the adjacency matrix in Section 3.1 is a technique used in Graph Convolutional Networks (GCNs) to prevent gradient vanishing or imploding and thus improve the stability and performance of the model. Symmetric normalization involves normalizing the adjacency matrix by the product of the square roots of the degree matrices. Thus, the normalized adjacency matrix of the directed graph would still be asymmetric.
> > 2.	Re-estimating contributions after edge selection: Thank you for your valuable suggestion.As shown in Figure 9 and 10, the prediction with the selected subgraph are very close to the prediction outcome of the original graph. Thus, we postulate that the feature contribution may not vary significantly.
> > 3.	Link function g in Eqn. 1: A link function is used to relate the predictor to the expected value of the response variable in a generalized linear model (GLM) and a generalized additive model. The link function provides a way to model the relationship between the predictor variables and the response variable when the response variable has a non-normal distribution, such as binary or count data.
> > 4.	Compact subgraph selection and classification accuracy: We appreciate your insightful comments. Figures 9 and 10 show that the predictions using the selected subgraph by the proposed approach are more similar to those using the entire graph under different sparsity levels compared with other methods.
> > We appreciate your valuable feedbacks, which helped us improve the quality and clarity of our work. We incorporated the requested changes in our revised manuscript and address your concerns. Thank you for your time and constructive review.

---

### Review · Reviewer_9b6w · 2023-06-27

**Summary Of Contributions:**

Interpretability of Graph Neural Networks (GNNs) is an important area of focus for the graph machine learning research community. This paper aims to provide a new method leveraging the Generalized Additive Model (GAM). The paper proposes to split the output into two parts - 1] Global level feature importances and 2] Edge attribution for local explanations. Both of these are modeled along the lines of GAM. For Global level feature importances, labels are directly modeled as a combination of feature-wise GNNs. In the edge attribution case, a k-hop neighborhood is taken around the target node and random edges are masked out. The inverted mask is used as feature vector, where the mask considers both the directions, instead of being undirected. A GAM is used to learn and predict the output given by the model under the said mask. The edges that explain the output variations are then identified by observing which of the features got higher weight.

**Audience:**

Yes

**Claims And Evidence:**

No

**Requested Changes:**

1. **Missing Motivation**: What are the issues with the prior approaches, that this particular method is trying to mitigate? And how is it mitigated by the proposed approach?
2. **Model Choice Explanation**: Why are the models designed the way they are? Why should $g(E[Y|X,A]) = \mu + \sum_{j}f_j(X_j, A)$ be even true and a reasonable modeling choice for feature importance? For local interpretation, why is a boolean edge vectors a sufficient feature vector to predict the output variations? It is claimed that modeling the directions is beneficial even in undirected graph. Why is that the case, when the given GNN model itself is operating on an undirected assumption?
3. **Clarity**: Why is Global feature importance not taking the learnt model into account? If it is taking into account, it is not clear in the writing, where is it being taken into account? If it is not taking into account, then is it something like feature selection problem being solved, in which case, this is not related to interpretation? The experiment section is not organized properly, it was very hard to understand what all things were analyzed and evaluated. Not to mention, certain metrics like fidelity are defined differently from what prior works have done and this definition is only clarified in a caption of a table which was hard to find.


**Strengths And Weaknesses:**

# Strengths
1. **Simplicity**: A very simple approach

# Weaknesses
1. **Missing Motivation**: Clear motivation for the approach and modeling is missing.
2. **Model Choice Explanation**: The modeling choices are not clearly explained.
3. **Clarity**: Some aspects of the model working are not clear.

---

> ### Author Response · Authors · 2023-07-28
>
> Dear Reviewer 9b6w,
>
> Thank you for your valuable comments and suggestions on our paper. We appreciate your thorough review and have carefully considered each of your points. Below, we provide a response addressing the raised concerns and requested changes.
>
> 1.	Missing Motivation:
> The paper's motivation is to address the limitations of existing post-hoc explanation tools for interpreting Graph Neural Networks (GNNs). These existing tools provide approximations of the original GNNs and may only partially capture the underlying patterns and structures. Meanwhile, these approaches often fail to provide a global view of the model. Our proposed method addresses this limitation by providing a unified framework that can capture both global and local feature importance. By leveraging the Generalized Additive Model (GAM), we can obtain a comprehensive global understanding of how different features contribute to the predictions. As the nodes in a graph usually have varied neighboring topology, we need a fine-grained local explanation to understand the relationships between the target node and its corresponding neighbor structures. This motivates our approach to enhance interpretability in GNNs by providing a holistic perspective on the importance of features and structures. We have further elaborated our motivation in the last paragraph of the introduction section.
>
> 2.	Model Choice Explanation:
> The generalized additive model (GAM) is a classic interpretable statistical machine learning model that extends linear models with nonparametric ridge functions. Ridge functions can be fitted by tree ensembles or re-parametrized using neural network architectures. The GAM is known to be flexible due to its nonparametric nature as well as interpretable because it is an extension of linear regression. Therefore, the GAM is an ideal choice as a statistical model to explain the black-box. We use subnetworks replace each ridge function, allowing for joint optimization using modern training techniques. In this paper, to leverage the intrinsic explainability of this family of glass-box models, we also make restrictions on the structure of graph neural networks. The model learns the linear combination of the feature networks that each attend to a single node feature with the graph structure. The global contribution of each single node feature can be easily visualized by plotting the feature network output against the feature values. As shown in Section 4.4, GAM-GCN preserves the expressive power of the black-box models in addition to the intrinsic interpretability. Further, GAM-GCN can also be used as a surrogate model to model the behavior of the black-box GNNs. In this way, we could show the contributions of each individual features at each possible value with respect to the prediction.
>
> 3.	Why modeling the directions?
> We propose the edge attribution algorithm to measure the effects of the edges. Empirically, we found that considering the directions when modeling edge attribution results in better performance when fitting the output variation than those modeling the undirected edges. Then, the importance of the undirected edge is obtained by averaging the edge coefficients in two directions, as illustrated in Equation 6. As shown in Table 4 in the updated manuscript, it is observed that our proposed local interpretation method with the simple linear model for neighboring edges generally outperforms the benchmarks on synthetic node classification datasets. Meanwhile, in Figures 9 and 10, we have shown that the predictions using the selected subgraph by the proposed approach are more similar to those using the entire graph under different sparsity levels compared with other methods.
>
> 4.	Clarity:
>
> “Why is Global feature importance not taking the learnt model into account? If it is taking into account, it is not clear in the writing, where is it being taken into account? If it is not taking into account, then is it something like feature selection problem being solved, in which case, this is not related to interpretation?”
> Please see our response in Q2.
>
> Regarding the organization of the experiment section, we have thoroughly revised the section to improve its clarity and ensure that the analyzed and evaluated aspects are presented in a more organized manner. We also acknowledge that the definition of fidelity follows the definition of Fidelity+ defined by Yuan et al. (2023). We provided a more explicit definition in the paper's main text to avoid any ambiguity in Eqn. 5 on Page 10.
>
> H. Yuan, H. Yu, S. Gui and S. Ji, "Explainability in Graph Neural Networks: A Taxonomic Survey" in IEEE Transactions on Pattern Analysis & Machine Intelligence, vol. 45, no. 05, pp. 5782-5799, 2023. doi: 10.1109/TPAMI.2022.3204236
>
> Thank you once again for your feedbacks. We believe that addressing these concerns has significantly improved the quality of our paper. We made the necessary revisions to address all your points.

---

### Review · Reviewer_KzVW · 2023-07-15

**Summary Of Contributions:**

The paper proposes to cast an interpretable GNN model as a generalized additive model which composes of the node's effects (which depend on nodal features) along with edge contributions. The proposed model is described in section 3. For a graph G = (V, E), X \in X^{n x d} are the d dimensional features associated with n nodes. Y denotes the response associated with each node. A denotes the adjacency matrix and \tilde{A} a version with self-loops allowed. Recall that the GAM model composes of d ridge functions which are combined linearly and the added with an intercept term. This is modified for the ridge functions to take as inputs X_j and A. Using such a formulation the overall importance of each node is measured by the mean absolute score, such that for all features it sums to1. This model is trained according to whatever the downstream task is. For edge attribution is described in algorithm 1. The associated surrogate is a linear model that uses the generated binary vectors to predict the outcome (why is this different from Y?). The edge attribution procedure is used select a compact subgraph that explains the prediction for a specific node. This selected subgraph is then visualized to show explainability. Both nodes and edges can be removed at randomly to arrive at a more general explainable model. Synthetic and real world experiments are used to argue for the applicability of the proposed methods. The proposed model is compared to some standard GNNs and explanation methods and compared in terms precision and recall etc.

**Audience:**

Yes

**Claims And Evidence:**

No

**Requested Changes:**

See above. Some additional comments for clarification.

Minor comments (issues such as this are present throughout the paper):
- Page 1: "The graph neural network (GNN) is a relatively new type of neural network that has drawn much attention in recent years (Zhou et al., 2020)." It is not a new type of neural network and has been around since the 90s. Also, please cite some notable papers in the GNN literature over here rather than a recent survey which doesn't have much to do with the development of GNNs.
- Page 1 "Without preprocessing, unstructured graph data can be directly modeled by GNNs" --> how it is unstructured if it is modeled as a graph? Please clarify what you mean.
- Page 2: "draws extensive attentions that GAM can also be reparametrized using neural network architectures" --> has drawn extensive attention? Please reword for grammatical correctness.
- Page 2: "Interpretable machine learning is an emerging topic in both industries and academia" --> both industry and academia. Again, this line is framed generally about interpretable machine learning, but the following line then proceeds as "so that further improvements on GNNs
can be achieved. In general, there are two types of interpretability" and switches to GNNs. Please reword to keep it flowing.
- Page 3: "The number of edges in the subgraph is marked as M. In addition, the adjacency matrix is denoted by A." --> Please use consistent notation. It seems like upper case letters are used to denote sets (V, E etc), matrices (A), and also for cardinality (M). Please use the same typeface for the same type of mathematical object.

**Strengths And Weaknesses:**

- The general idea proposed here is both natural and potentially useful.

Weaknesses:
- I found the paper very hard to read. It often has serious clarity issues. Even in the abstract, I couldn't quite get what the authors were trying to say (except the GAM part, which is straightforward) till I read the introduction many times. I would request the authors to do a thorough rewrite of the paper -- starting with the abstract. For example I can only guess what "Further, the effects of neighboring edges are measured by edge perturbation and surrogate linear modeling, and the most important subgraph can be selected" or "The inherent
interpretability of GAM and the expressive power of GCN are preserved and naturally connected." mean. It was only after reading the intro, I kind of got what it was implying.
- In section 3 it was not clear to me how the node attribution and feature attribution were connected together. Either this is missed to explain, or there is a notation mismatch. What is the outcome O for edge is never explained. With this it took looking at the experiments to understand what was going on.
- The proposed model is a kind of surrogate model which learns an inherently (arguably) interpretable GNN. However, the initial motivation seemed to suggest that we would be making a general GNN model explainable. The explainability here is in terms of the visualization of the node and edge contributions. For complex graphs, it would help to present more examples. Consider molecular property prediction for simplicity. It would be interesting to show that a subgraph picked up is a functional group responsible for predicted property. For large graphs and per node classification I am not convinced (at least by how the author have argued for it) that the method is helping explanation. The AUC and precision recall curves are only telling us that the model is competitive with existing graph NNs. But the main contribution here is of explainability. I am not sure that the experiments (except the language example) is really making the case for it. I found the example graphs attached hard to interpret, for instance. Perhaps they should be real graphs, let's say small world networks, and then perhaps the visualizations would be more intuitive.

---

> ### Author Response · Authors · 2023-07-28
>
> Dear Reviewer KzVW,
>
> Thank you for your thoughtful review and valuable feedback on our paper. We appreciate your insights and understand the areas where the paper needs improvement. Below, we provide a response to your concerns and address the issues raised in your review.
>
> Sorry that we did not convey our idea clearly. In the revised manuscript, we rewrote the introduction, and experiment sections to make them more accessible and easier to understand to better convey the motivations and contributions of our proposed method. These existing tools provide approximations of the original GNNs and may only partially capture the underlying patterns and structures. Our proposed method provides a unified framework that can capture both global and local feature importance. By leveraging the Generalized Additive Model (GAM), we can obtain a comprehensive global understanding of how different features contribute to the predictions. This motivates our approach to enhance interpretability in GNNs by providing a holistic perspective on the importance of features and structures. As the nodes in a graph usually have varied neighboring topology, we need a fine-grained local explanation to understand the relationships between the target node and its corresponding neighbor structures. Thus, we propose to use the edge attribution methods to provide local explanations for each node as the explanation of local graph structures for node classification task.
>
> As shown in Section 3.3.1, we pass the perturbed graph to the fitted GNN to get the predicted outcome (O). By repeatedly perturbing the input graph and calculating the corresponding prediction, we can run a linear regression model on the generated data, where the variables are the absence or presence of the edges, and the target is the predicted output O we gained above.
>
> There are many different ways to explain a black-box. One popular way is to use a surrogate model that is interpretable to explain the black-box. The GAM has been well studied in statistics and has good interpretations as it is an extension of linear regression in a nonparametric setting. Our work conducts explainability in terms of the visualization of the node and edge contributions. We emphasize our work is one way to explain the black-box.
>
> We thank you for your valuable suggestions and further addressed the minor issues throughout the paper, correcting grammar, and providing more appropriate citations for the history of GNNs and improving the consistency in our notations to avoid confusion.
>
> Thank you again for your detailed and insightful review. Your feedback significantly helps us improve the quality and clarity of our work. We are committed to addressing all the concerns raised and making the necessary revisions to present a more coherent and impactful manuscript.

---

### Decision · Action_Editors · 2023-08-23

**Recommendation:** Reject

**Comment:**

Reviewers generally agreed that the problem tackled by this submission is important and of high relevance to the TMLR community.  They also appreciated the simplicity and straightforwardness of the proposed approach.  However, significant concerns were raised with regards to clarity of presentation, justification behind choices made, and a clear definition of the problem with existing methods that the proposed approach aims to solve.  In light of these concerns I cannot recommend acceptance of the submission in its current form.  I encourage the authors to take the reviewers' comments into account for improving the manuscript.

**Audience:**

The general topic of this submission --- interpretability of GNNs --- is highly relevant to the TMLR community.

**Claims And Evidence:**

Evidence supporting claims is provided, but reviewers felt that there are gaps warranting a major revision --- see general comment.

**Resubmission Of Major Revision:**

The authors may consider submitting a major revision at a later time.